# Gradient GA: Gradient Genetic Algorithm For Drug Molecular Design

**Debadyuti Mukherjee**[*]                                    *mukher83@purdue.edu*
*Department of Computer Science*
*Purdue University*

**Chris Zhuang**[*]                                           *zhuang80@purdue.edu*
*Department of Computer Science*
*Purdue University*

**Yingzhou Lu**                                               *lyz66@stanford.edu*
*Stanford Medicine School*
*Standford University*

**Tianfan Fu**                                                *fut2@rpi.edu*
*Department of Computer Science*
*Rensselaer Polytechnic Institute*

**Ruqi Zhang**                                                *ruqiz@purdue.edu*
*Department of Computer Science*
*Purdue University*

**Reviewed on OpenReview:** *https://openreview.net/forum?id=kFKcktAeEG*

## Abstract

Molecular discovery has brought great benefit to the chemical industry. Various molecular design techniques have been developed to identify molecules with desirable properties. Traditional optimization methods, such as genetic algorithms, continue to achieve state-of-the-art results across various molecular design benchmarks. However, these techniques rely solely on undirected random exploration, which hinders both the quality of the final solution and the convergence speed. To address this limitation, we propose a novel approach called *Gradient Genetic Algorithm* (Gradient GA), which incorporates gradient information from the objective function into genetic algorithms. Instead of random exploration, each proposed sample iteratively progresses toward an optimal solution by following the gradient direction. We achieve this by designing a differentiable objective function parameterized by a neural network and utilizing the Discrete Langevin Proposal to enable gradient guidance in discrete molecular spaces. Experimental results demonstrate that our method significantly improves both convergence speed and solution quality, outperforming cutting-edge techniques. The proposed method has shown up to a 25% improvement in the Top 10 score over the vanilla genetic algorithm. The code is available at `https://github.com/debadyuti23/GradientGA`.

## 1 Introduction

Designing molecules with desirable biological and chemical properties has become a demanding research topic since its outcome can benefit various domains, such as drug discovery (Huang et al., 2022), material design (Yang et al., 2017), etc. However, a limited number of molecules can be tested in real-life laboratories

---

[*]Equal contribution. First authorship determined by rolling a dice.

(Altae-Tran et al., 2017) and clinical trials (Chen et al., 2024b;a). Therefore, numerous effective techniques for molecule discovery have been proposed to discover favorable molecules throughout the vast sample space.

Some evolutionary algorithms, such as molecular graph-based genetic algorithm (Graph GA) (Jensen, 2019a), remain strong baselines, often outperforming recently proposed machine learning-based algorithms (Huang et al., 2021; Gao et al., 2022b). Genetic algorithms are cheap, easy to implement, and often regarded as simple baselines for molecular discovery. However, key GA operators, such as selection, crossover, and mutation, are random and do not use knowledge of objective functions. Given the vast molecular search space, this random walk approach is like searching for a needle in a haystack. As a result, GA tends to converge slowly, and its final performance can be unstable.

To address this issue, we introduce a novel molecule design method, *Gradient Genetic Algorithm* (Gradient GA), which leverages gradient information to navigate chemical space efficiently. First, we learn a differentiable objective function using a graph neural network (GNN) (Scarselli et al., 2009), which maps the graph-structured information of molecules to vector embeddings. We then apply the Discrete Langevin Proposal (DLP) (Zhang et al., 2022) to incorporate gradient information from this objective, enabling more informed exploration in the discrete molecular space. Our main contributions are summarized as follows.

- We introduce Gradient GA, a gradient-based genetic algorithm for more informative and effective exploration in molecular spaces, mitigating random-walk behavior in genetic algorithms. To the best of our knowledge, this is the first method to leverage gradient information within a genetic algorithm framework.

- To enable a differentiable objective function for discrete molecular graphs, we use a graph neural network as a property predictor to approximate non-differentiable objectives. This allows us to compute gradients by taking derivatives of NN-parameterized objectives with respect to the vector embeddings.

- The experimental results demonstrate that the proposed method achieves a significant and consistent improvement over a number of cutting-edge approaches (e.g., Graph-GA, SMILES-GA). For example, achieving an improvement of up to 25% over the traditional GA when optimizing the mestranol similarity property.

## 2 Related Work

**AI-aided Drug Molecular Design.** Current AI-aided drug molecular design techniques can be primarily classified into two categories: deep generative models and combinatorial optimization methods.

(I) Deep Generative Models (DGMs) learn the distribution of general molecular structures using deep networks, enabling the generation of molecules by sampling from the learned distribution. Typical algorithms include Variational Autoencoders (VAEs), Generative Adversarial Networks (GANs), energy-based models, and flow-based models (Gómez-Bombarelli et al., 2018; Jin et al., 2018; De Cao & Kipf, 2018; Segler et al., 2018; Fu et al., 2020; Honda et al., 2019; Madhawa et al., 2019; Liu et al., 2021; Fu & Sun, 2022; Chen et al., 2024b; Bagal et al., 2021). However, these approaches often require a smooth and discriminative latent space, necessitating careful network architecture design and well-distributed datasets. This requirement can be restrictive in certain scenarios, such as multi-objective optimization. Furthermore, since DGMs learn the distribution of reference data, their ability to explore diverse chemical space is relatively limited, as demonstrated by recent molecular optimization benchmarks (Brown et al., 2019; Huang et al., 2021; Gao et al., 2022b).

(II) On the other hand, combinatorial optimization methods directly search the discrete chemical space, mainly including deep reinforcement learning (You et al., 2018; Zhou et al., 2019; Jin et al., 2020; Gottipati et al., 2020), evolutionary learning methods (Nigam et al., 2020; Jensen, 2019b; Fu et al., 2022a), and sampling methods (Xie et al., 2021; Fu et al., 2021). Specifically, Jensen (2019a) has proposed a molecular graph-based genetic algorithm. In drug discovery, this algorithm samples two parent molecules and generates child molecules by combining fragments of the parents, with a probability of random mutations occurring in

the offspring. The population is then refined by selecting the highest-scoring molecules. Also, Differentiable Scaffolding Tree (DST) (Fu et al., 2022b) differentiates the structure of discrete molecules and enables gradient computation for efficient optimization; Xie et al. (2021) uses the Markov chain Monte Carlo method (MCMC) to sample potential molecules. Each sampled molecule forms a Markov chain, modeled as a chemical transformation of the previous sample. This transformation occurs through one of two possible actions: (i) the addition of molecular fragments or (ii) the removal of a chemical bond. However, both MARS and DST rely on a single selected molecule and fail to incorporate other molecules' properties, unlike GraphGA, hurting their top performance. This motivates us to propose an AI-guided genetic algorithm.

**Discrete Sampling.** Many applications involve discrete data spaces, such as molecular, text, and tabular data. Gibbs sampling has long been the standard method for discrete sampling. However, because Gibbs sampling updates only one variable at a time, it often suffers from slow convergence. To address this, various improvements to Gibbs sampling have been proposed. Titsias & Yau (2017) introduces auxiliary variables to enable block updates within the Gibbs sampler. Nishimura & Suchard (2023) reformulates the sample space to simplify the sampling process by using prior preconditioning and conjugate gradient techniques. Recently, a growing body of work has explored the use of gradient information to improve sampling in discrete spaces (Grathwohl et al., 2021; Sun et al., 2022; Pynadath et al., 2024). Among these, the Discrete Langevin Proposal (DLP) (Zhang et al., 2022) stands out as an analog of the Langevin dynamics adapted to discrete spaces. DLP not only utilizes gradient information but also updates all variables simultaneously at each step, leading to better efficiency.

## 3 Preliminaries

### 3.1 Discrete Langevin Proposal

Suppose that the target distribution is $\pi(v) \propto \exp(U(v))$, where $U(\cdot)$ is the energy function, $v$ is a $n$-dimensional variable in the space $\mathbb{R}^n$. Langevin Dynamics samples from $\pi$ by iteratively updating $v$ as follows:

$$v' = v + \frac{\alpha}{2}\nabla U(v) + \sqrt{\alpha}\varepsilon, \quad \varepsilon \sim \mathcal{N}(0, I_{n \times n}), \tag{1}$$

where $\alpha$ is the step size; $I_{n \times n}$ is $n$-dimensional identity matrix; $\mathcal{N}(\cdot, \cdot)$ denotes high-dimensional normal distribution.

From Equation 1, the probability of selecting $v'$ from $V$, i.e., $p(v'|v)$, can be written as

$$p(v'|v) \propto \exp\left(-\frac{1}{2\alpha}||v' - v - \frac{\alpha}{2}\nabla U(v)||_2^2\right), \tag{2}$$

The distribution $p(v'|v)$ has its mean shifted from $v$ toward the optimum due to the gradient term. Therefore, high-probability samples from $p(v'|v)$ will be closer to the optimum compared to $v$.

To extend Langevin Dynamics to discrete space, Zhang et al. (2022) has suggested the following proposal for the discrete sample space $S$:

$$p(v'|v) = \frac{\exp\left(-\frac{1}{2\alpha}||v' - v - \frac{\alpha}{2}\nabla U(v)||_2^2\right)}{\sum_{v'' \in S}\left[\exp\left(-\frac{1}{2\alpha}||v'' - v - \frac{\alpha}{2}\nabla U(v)||_2^2\right)\right]}. \tag{3}$$

### 3.2 Genetic Algorithm

The genetic algorithm (GA) is a traditional combinatorial optimization method motivated by natural selection and biological evolution processes. Unlike neural network-based methods, GA does not have learnable parameters, is usually easy to implement and tune, and bypasses the overfitting issue. Specifically, a GA process starts by randomly sampling a *population* of candidates. In drug design, for example, the candidates can be drug molecules. In the $t$-th *generation* (iteration), given the population of candidates, GA follows three key steps.

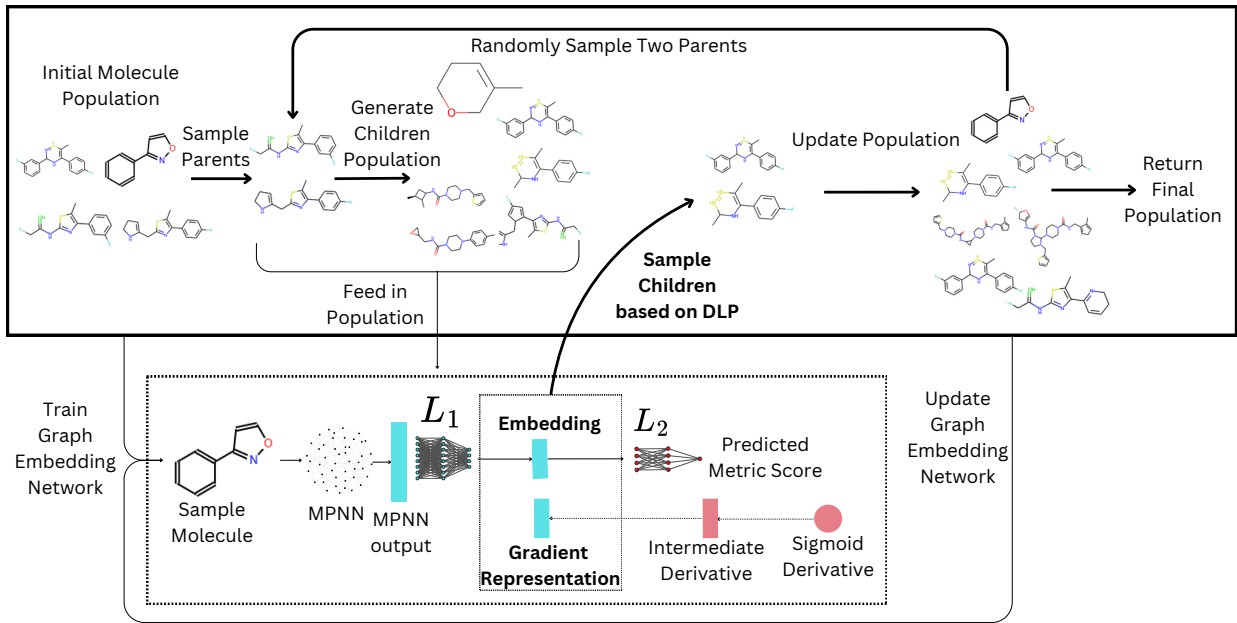

Figure 1: Gradient GA pipeline.

1. **Crossover**, also called recombination, exchanges the structure of two parents to generate new children. Specifically, two parents are randomly selected from the population, and their molecular structures are partially swapped to create two child molecules. The crossover operators are carried out multiple times independently, and the generated children are added to the offspring set $\mathcal{S}^{(t)}$.

2. **Mutation** operates on a single parent molecule (randomly selected from the population) and modifies its structure slightly via randomly selecting a substructure and flipping it to a new substructure different from its original State. Like the crossover, the mutation is conducted multiple times independently, and the resulting offspring are retained in the offspring set $\mathcal{S}^{(t)}$.

3. **Evolution**. Given a population of molecules $\mathcal{S}^{(t)}$ at the $t$-th generation, we generate an offspring pool by applying crossover and mutation operations. Molecules with undesirable properties (e.g., poor solubility or high toxicity in drug discovery) are filtered out, and the top $k$ candidates are selected to form the next generation population $\mathcal{S}^{(t+1)}$.

## 4 Methodology: Gradient Genetic Algorithm

**Overview.** In this section, we introduce Gradient Genetic Algorithm (Gradient GA). Section 4.1 begins by formulating the molecular design problem. Then, Section 4.2 describes how to derive gradients by projecting discrete molecular graphs into a continuous embedding space. Next, Section 4.3 explains how to use the embedding-derived gradients to efficiently explore the molecular space. For clarity, all mathematical notations are listed and explained in Table 1. The overall pipeline is illustrated in Figure 1, and the corresponding pseudocode is provided in Algorithm 1.

### 4.1 Formulation: Molecular Design

Drug molecular design aims at identifying novel molecules with desirable pharmaceutical properties, which are evaluated by *oracle*. Oracles serve as objective functions in molecular optimization tasks, formally defined as follows.

**Definition 4.1** (Oracle). Oracle $\mathcal{O}(\cdot) : \mathcal{Q} \to \mathbb{R}$ is a black-box function that evaluates certain physical, chemical, or biological properties of a molecule $X$ and yields the ground-truth property $\mathcal{O}(X)$.

Table 1: Mathematical notations and their explanations.

| Notations | Explanations |
|---|---|
| $\mathcal{O}$ | oracle function |
| $\mathcal{Q}$ | molecular space |
| $\mathcal{M}$ | complete MPNN-based model for oracle prediction |
| $G(\cdot)$ | graph representation |
| $L$ | multiple-layer perceptron (MLP) |
| $\pi(\cdot)$ | target distribution (normalized score) |
| $U(\cdot)$ | energy function |
| $D$ | population, a set of molecules. |

In drug discovery, all the oracles can be categorized into two classes based on their accessibility: computational and experimental (wet-lab). Experimental oracles, e.g., *in vivo* experiment, typically require wet-lab experiments to evaluate, which are too expensive and time-consuming. Following most machine learning-aided drug discovery papers, we focus on computational oracles that are easy to evaluate *in silico*, such as molecular docking and the quantitative estimate of drug-likeness (QED) (Bickerton et al., 2012). In real-world drug discovery scenarios, the cost of acquiring oracle evaluations is typically significant and cannot be overlooked. Mathematically, the drug molecular design problem (a.k.a. molecular optimization) can be formulated as

$$\arg\max_{X \in \mathcal{Q}} \mathcal{O}(X), \tag{4}$$

where $X$ is a molecule, $\mathcal{Q}$ denotes the whole molecular space, i.e., the set of all chemically valid molecules. The size of the whole molecular space is around $10^{60}$ (Bohacek et al., 1996) Following MARS (Xie et al., 2021), we regard $\mathcal{O}(\cdot)$ as an unnormalized probability distribution and introduce a vector embedding $v$ for each molecule $X$. The target distribution is then defined as $\pi(v) \propto \mathcal{O}(X)$.

**Gradient Definition in Molecular Design.** We now define the gradient information used to guide exploration in molecular spaces. To apply (discrete) Langevin dynamics, we need the gradient of the energy function. Since $\pi(v) \propto \exp(U(v))$, by the chain rule, we have

$$\nabla U(v) = \frac{\nabla \pi(v)}{\pi(v)}. \tag{5}$$

Given that $\pi(v) \propto \mathcal{O}(X)$, this leads to

$$\nabla U(v) = \frac{\nabla \mathcal{O}(X)}{\mathcal{O}(X)} = \frac{\nabla f(v)}{\mathcal{O}(X)}, \tag{6}$$

where $f$ is a differentiable function that approximates the oracle $\mathcal{O}$, i.e., $f(v) = \mathcal{O}(X)$ for all $X \in \mathcal{Q}$. We will discuss how to obtain both $f$ and $v$ in the next section.

It is also possible to define $U(v) = \mathcal{O}(X)$ and $\nabla U(v) = \nabla f(v)$. However, this formulation does not incorporate the oracle value into the gradient. Our approach includes $\mathcal{O}(X)$ in the denominator, effectively playing the role of adaptive step sizes. We found that this formulation leads to better performance. An empirical comparison is provided in the Appendix C.

## 4.2 Gradient Computation

Implementing gradient-based methods in molecular discovery is a challenging task due to two primary obstacles: (1) representing sample molecules in a vector format suitable for gradient-based methods, and (2) establishing a differentiable relationship between the probability distribution and the vector representation.

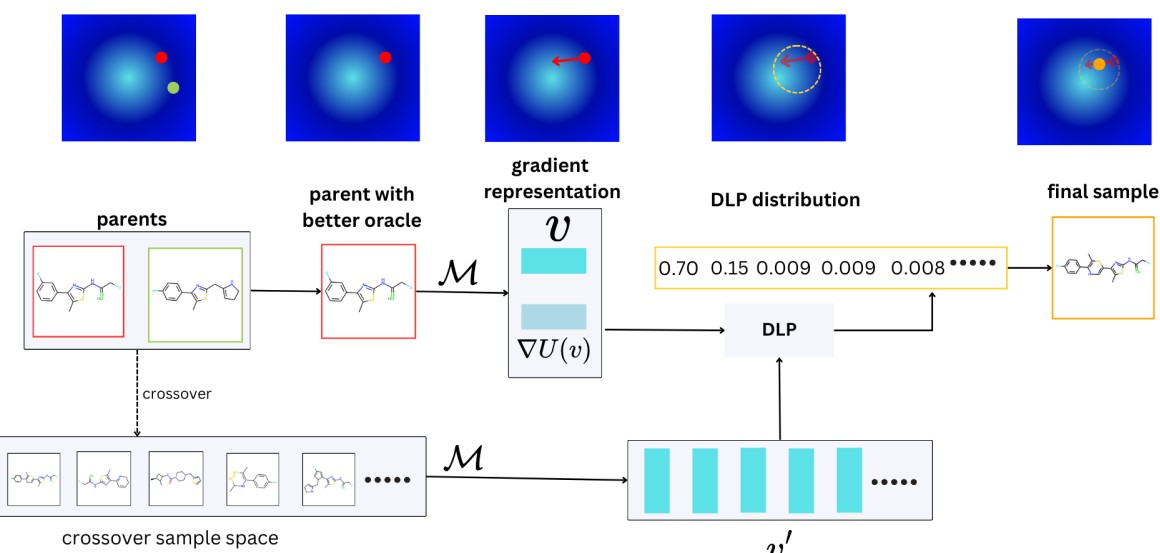

Figure 2: Overview of DLP-based sampling procedure in Gradient GA, illustrating how the sampled molecule moves toward the optimum.

**Finding the Embedding.** Most gradient-based methods use fixed-length vectors, while molecular data are often represented as graph structures. Therefore, an efficient mapping function is needed to convert molecular graphs into fixed-length embeddings. However, directly transforming a graph into an embedding using hard-coded functions, such as Morgan's fingerprint, can cause unnecessary aggregation and information loss (Chang et al., 2019). To overcome this issue, we use the Message Passing Neural Network (MPNN) (Gilmer et al., 2017), which is a State-of-the-art approach for molecular activity prediction. We adopt the MPNN architecture from Xie et al. (2021), which consists of (1) a simple neural network for message passing between neighboring nodes and (2) a Gated Recurrent Unit (GRU) (Cho, 2014) for updating node representations. For the readout function, we use Set2Set (Vinyals et al., 2015), which is particularly effective for isomorphic graphs due to its set-invariant property. To reduce the dimensionality of the Set2Set output, we introduce a two-layer multilayer perceptron (MLP) $L_1$, which compresses the output dimension from $2n$ to $n$.

We define the graph representation function, which converts molecular data into graph data (nodes representing atomic features and edges representing bond features), as $G(\cdot)$. Consequently, the output of $L_1$ serves as the embedding $v$ for the molecular graph $G_X = G(X)$, as shown below.

$$h = \text{MPNN}(G_X), v = L_1(h). \tag{7}$$

**Finding the Gradient.** Similar to the vector embedding problem, we can learn the differentiable approximation $f(\cdot)$ in Equation 6 using an MLP-based architecture. After the first two-layer MLP $L_1$, we introduce another two-layer MLP $L_2$, which produces a scalar output $\widehat{y}$. The gradient $\nabla f(\cdot)$ can then be computed through backpropagation, as illustrated in Figure 1. Our objective is to make $\widehat{y}$ approximate the behavior of the oracle function $\mathcal{O}$ so that the entire architecture (from the MPNN to $L_2$), denoted by $\mathcal{M}$, effectively learns both the embedding and the differentiable objective function. To introduce nonlinearity, we apply LeakyReLU and sigmoid activation functions before and after $L_2$, respectively. With $v$ and $\nabla U(\cdot)$ now properly defined for Equation 3, we proceed to describe the training procedure. We train the model $\mathcal{M}$ to fit the oracle function $\mathcal{O}$ using the initial molecule population $D$. The loss function is defined as the mean squared error (MSE):

$$\mathcal{L} = \frac{1}{|D|} \sum_{X \in D} ||\mathcal{M}(G_X) - \mathcal{O}(X)||^2.$$

The initial molecule population is randomly sampled from drug-like molecule databases, such as ZINC (Sterling & Irwin, 2015). During the molecular optimization process, we continuously expand the training set by adding newly generated molecules that surpass a threshold criterion $T$. This provides additional information about the distribution of molecules and allows us to retrain the model $\mathcal{M}$. To ensure that the approximation from $\mathcal{M}$ is accurate, we provide empirical results of the performance using a validation set in Appendix D

### 4.3 Iterative Sampling

We propose a sampling technique inspired by both Graph GA (Jensen, 2019a) and DLP (Zhang et al., 2022). The workflow for each iteration is illustrated in Figure 2. Similar to Graph GA, we begin by selecting parent molecules based on their scores $\mathcal{O}(\cdot)$ and generating child molecules through crossover. We define the sample space $S$ for DLP as the set of all possible crossovers between the selected parents $d_1$ and $d_2$ ($\in D$). Ideally, both parents should be considered as current samples. However, since DLP is designed to use a single sample, we aggregate the information from both parents into a single embedding $v$ and gradient $\nabla U(v)$ using the following equation:

$$\{v, \nabla U(v)\} = \sum_{i=1,2} w_i \cdot \{v_i, \nabla U(v_i)\}, \tag{8}$$

where $\{v_i, \nabla U(v_i)\}$ represents the embedding and gradient information for parent $d_i$. Empirically, we found that a simple strategy of using only the best parent as the current sample works well: $w_i = 1$ if $i = \arg\max(\mathcal{O}(d_i))$, and $w_i = 0$ otherwise; we have provided a comparison of the performance when averaging the parents in Appendix E. DLP updates the embedding $v'$ by moving it closer to the optimum, guided by the gradient information. In the final step, DLP generates the next sample set $D'$ of fixed size $k$ from the sample space $S$. Following the Graph GA approach, each molecule in $D'$ is mutated. Before the next iteration begins, we update both the population and the model. The population $D$ is refreshed by selecting the top $|D|$ molecules based on their oracle scores from the combined set $\{D, D'\}$. To further enhance the graph embedding model $\mathcal{M}$'s understanding of the target molecule distribution $\pi(\cdot)$, we retrain $\mathcal{M}$ using a training set $D''$, which is updated in each iteration according to the following rule:

$$D'' = D'' \cup \{d \mid d \in D' \text{ and } T(d)\}, \tag{9}$$

where $T(d)$ is a threshold criterion for adding new samples to the training set. The complete Gradient GA workflow is detailed in Algorithm 1.

## 5 Experiments

### 5.1 Experimental Setup

**Baseline Methods.** We use the practical molecule optimization (PMO) benchmark (Gao et al., 2022a) as our code base to compare results with the state-of-the-art methods. We select (1) genetic algorithm, including Graph GA (molecular graph-level genetic algorithm) method (Jensen, 2019b) and SMILES GA (SMILES string-level genetic algorithm), (2) sampling-based methods, including MIMOSA (Multi-constraint Molecule Sampling) (Fu et al., 2021), MARS (Markov Molecular Sampling) (Xie et al., 2021), and (3) gradient-based method, DST (Differentiable Scaffolding Tree) (Fu et al., 2022b). We perform a maximum of 2500 oracle calls for each method with early-stopping being enabled when negligible performance improvement occurs a certain number of times.

**Dataset.** For all methods, we use the ZINC 250K database (Irwin et al., 2012) to select the initial molecule population, extract chemical fragments, and perform pretraining. ZINC is a free database of commercially available compounds for virtual screening.

**Evaluation Metrics.** We consider the following evaluation metrics to assess the quality of generated molecules.

1. **Average Top-$K$** ($K = 10$) is the top-$K$ average property value, which measures the algorithm's optimization ability. We limit the number of oracle calls to 2,500 to mimic the real experimental setup, though we expect methods to optimize well within hundreds of calls.

---

**Algorithm 1** Gradient Genetic Algorithm

---

**Input**: oracle function $\mathcal{O}$, step size $\alpha$, retrainable threshold $\tau$
Initialize $D \leftarrow$ original population
Initialize new molecule set $D' \leftarrow \{\}$
Train predictive model $\mathcal{M}$ using
$\{(G(d), \mathcal{O}(d)) \quad \forall d \in D\}$
Initialize retrained molecule set $D'' \leftarrow \{\}$
**for** $t = 1, 2, \ldots$ **do**
    $p(d) \propto \mathcal{O}(d) \quad \forall d \in D$
    Parent molecules $d_1, d_2 \sim p(d) \quad [\ d \in D]$
    Get parents' embedding $v_i$ for each $G(d_i)$ using Eq. 7
    Get parents' gradient $\nabla U(v_i)$ for each $d_i$ using Eq. 6
    Evaluate $v, \nabla U(v)$ using Eq. 8
    Get crossover set: $S \leftarrow \texttt{CROSSOVER}(d_1, d_2)$
    Get sampling probability $probs$ of $S$ using Eq. 3
    Evaluate sample set $D' \leftarrow \texttt{Sample}(S, probs, k)$
    Mutate each molecule in $D' \leftarrow \texttt{MUTATE}(D')$
    Update population $D \leftarrow \texttt{top\_oracle}(\{D, D_1\}, |D|)$
    Update training set $D''$ with $T$ using Eq. 9
    **if** $|D''| \geq \tau$ **then**
        Retrain model $\mathcal{M}$ with $\{(d, \mathcal{O}(d)) \quad \forall d \in D''\}$
    **end if**
    $D'' \leftarrow \{\}$ or $D'' \leftarrow D''$
**end for**

---

2. **AUC top-$K$** ($K = 1, 10, 100$). Following the setup of practical molecular optimization benchmark (Gao et al., 2022b), we assess both optimization ability and sample efficiency using the area under the curve (AUC) of top-$K$ average property value versus the number of oracle calls (*AUC top-K*) as the primary metric for measuring performance. Unlike using top-$K$ average property, AUC rewards methods that reach high values with fewer oracle calls. We use $K = 10$ as it is useful to identify a small number of distinct molecular candidates to progress to later stages of development. The reported values of AUC are scaled from min to max at $[0, 1]$.

3. **Diversity** of generated molecules is defined as the average pairwise Tanimoto distance between the Morgan fingerprints, formulated as

$$\text{diversity}(\mathcal{Z}) = 1 - \frac{1}{|\mathcal{Z}|(|\mathcal{Z}| - 1)} \sum_{\mathbf{z}_1, \mathbf{z}_2 \in \mathcal{Z}, \mathbf{z}_1 \neq \mathbf{z}_2} \text{sim}(\mathbf{z}_1, \mathbf{z}_2), \tag{10}$$

where $\mathcal{Z}$ is the set of generated molecules to evaluate and $\text{sim}(\mathbf{z}_1, \mathbf{z}_2)$ is the Tanimoto similarity between molecule $\mathbf{z}_1$ and $\mathbf{z}_2$. Diversity score ranges from 0 to 1, with higher values indicating greater diversity

4. **Synthetic accessibility (SA)** measures the difficulty of synthesizing the given molecule. The SA score values range from 1 to 10, where lower values indicate molecules that are easier to synthesize.

All these metrics can be calculated via the evaluation function in Therapeutics data commons (TDC) (Huang et al., 2021; 2022) [1].

**Implementation Details**. We select Mean Squared Error (MSE) and Adam (Kingma & Ba, 2014) with a learning rate of 0.001 as the loss function and optimizer, respectively, when training $\mathcal{M}$. The model is trained for 200 epochs for each round of training. The MPNN has 2 layers of convolution and GRU with 1 layer of Set2Set. There are $n = 16$ hidden features. The upper bound of the generated molecule $k$ is set to 70. The threshold criterion $\tau$ for a good score of a molecule is set as the maximum metric score $-0.001$. We

---

[1] https://tdcommons.ai/functions/data_evaluation/ and https://tdcommons.ai/functions/oracles/

have two experimental setups depending on the parameter $D''$. The first setup is to clear all the molecules that meet $\tau$ and retrain whenever there is an adequate amount. The second setup is to store the $D''$ and retrain at set oracle steps (default = 500 oracle calls). For simplicity, we take the setup of keeping $D''$ over each oracle call to have more consistent training at each iteration. We provide detailed tables of results for both setups in Appendix A. Both setups use the default Graph GA settings from PMO.

| Method | mestranol similarity | | | | amlodipine MPO | | | |
|---|---|---|---|---|---|---|---|---|
| | Average Top 10 | AUC Top 1 | AUC Top 10 | AUC Top 100 | Average Top 10 | AUC Top 1 | AUC Top 10 | AUC Top 100 |
| Gradient GA | **0.5130±0.0393** | **0.4433±0.0310** | **0.4082±0.0315** | **0.3534±0.0355** | **0.5667±0.0336** | **0.5614±0.0177** | **0.5176±0.0187** | 0.4658±0.0199 |
| Graph GA | 0.4452±0.0241 | 0.3556±0.0268 | 0.3208±0.0199 | 0.2717±0.0147 | 0.5605±0.0364 | 0.5067±0.0270 | 0.4734±0.0215 | 0.4152±0.0142 |
| SMILES GA | 0.2582±0.0097 | 0.3777±0.0381 | 0.3634±0.0352 | 0.3347±0.0279 | 0.4480±0.0161 | 0.5016±0.0156 | 0.4956±0.0143 | **0.4748±0.0158** |
| MIMOSA | 0.4262±0.0246 | 0.4162±0.0115 | 0.3619±0.0181 | 0.2887±0.0252 | 0.5245±0.0143 | 0.5431±0.0261 | 0.4953±0.0109 | 0.4436±0.0075 |
| MARS | 0.3411±0.0160 | 0.3760±0.0003 | 0.3215±0.0096 | 0.2523±0.0081 | 0.4843±0.0210 | 0.4812±0.0144 | 0.4583±0.0098 | 0.3816±0.0157 |
| DST | 0.4131±0.0179 | 0.4148±0.0323 | 0.3507±0.0088 | 0.2780±0.0029 | 0.5192±0.0122 | 0.5411±0.0303 | 0.4908±0.0115 | 0.4257±0.0044 |

| | perindopril MPO | | | | deco hop | | | |
|---|---|---|---|---|---|---|---|---|
| | Average Top 10 | AUC Top 1 | AUC Top 10 | AUC Top 100 | Average Top 10 | AUC Top 1 | AUC Top 10 | AUC Top 100 |
| Gradient GA | 0.4786±0.0257 | **0.4542±0.0164** | **0.4361±0.0176** | 0.3882±0.0193 | 0.6026±0.0053 | **0.5883±0.0032** | 0.5763±0.0050 | 0.5602±0.0053 |
| Graph GA | **0.4788±0.0067** | 0.4519±0.0055 | 0.4317±0.0045 | 0.3770±0.0049 | **0.6039±0.0043** | 0.5186±0.0037 | 0.5028±0.0032 | 0.4708±0.0033 |
| SMILES GA | 0.3698±0.0117 | 0.4346±0.0124 | 0.4271±0.0115 | **0.4065±0.0102** | 0.5548±0.0059 | 0.5862±0.0047 | **0.5817±0.0042** | **0.5733±0.0036** |
| MIMOSA | 0.4629±0.0176 | 0.4500±0.0144 | 0.4289±0.0116 | 0.3783±0.0085 | 0.6008±0.0053 | 0.5882±0.0061 | 0.5773±0.0035 | 0.5600±0.0021 |
| MARS | 0.4564±0.0167 | 0.4538±0.0087 | 0.4278±0.0065 | 0.3648±0.0042 | 0.5944±0.0070 | 0.5830±0.0227 | 0.5711±0.0301 | 0.5493±0.0421 |
| DST | 0.4615±0.0100 | 0.4530±0.0041 | 0.4210±0.0041 | 0.3564±0.0028 | 0.6034±0.0083 | 0.5860±0.0071 | 0.5721±0.0025 | 0.5518±0.0009 |

| | median1 | | | | isomers c9h10n2o2pf2cl | | | |
|---|---|---|---|---|---|---|---|---|
| | Average Top 10 | AUC Top 1 | AUC Top 10 | AUC Top 100 | Average Top 10 | AUC Top 1 | AUC Top 10 | AUC Top 100 |
| Gradient GA | **0.3033±0.0074** | **0.2581±0.0115** | **0.2298±0.0151** | **0.1906±0.0183** | 0.7783±0.0959 | 0.6628±0.0731 | 0.5444±0.0693 | 0.4033±0.0614 |
| Graph GA | 0.2599±0.0182 | 0.2315±0.0206 | 0.1959±0.0148 | 0.1442±0.0070 | 0.7222±0.1119 | 0.6648±0.0957 | 0.5436±0.0770 | 0.3891±0.0425 |
| SMILES GA | 0.1310±0.0172 | 0.1832±0.0281 | 0.1795±0.0272 | 0.1697±0.0251 | 0.3180±0.3583 | **0.8244±0.0848** | **0.7825±0.0752** | **0.7055±0.0668** |
| MIMOSA | 0.2391±0.0080 | 0.2271±0.0103 | 0.1969±0.0044 | 0.1537±0.0030 | **0.7866±0.0824** | 0.6965±0.0562 | 0.5949±0.0440 | 0.3965±0.0265 |
| MARS | 0.2094±0.0181 | 0.2239±0.0140 | 0.2019±0.0116 | 0.1671±0.0158 | 0.6639±0.1606 | 0.6751±0.1032 | 0.5909±0.1057 | 0.4424±0.1499 |
| DST | 0.2179±0.0162 | 0.2097±0.0086 | 0.1765±0.0021 | 0.1331±0.0024 | 0.6748±0.0304 | 0.6305±0.0435 | 0.4932±0.0216 | 0.2293±0.0093 |

Table 2: Comparison of Average Top 10, AUC Top 1, AUC Top 10, and AUC Top 100 with several GuacaMol objectives (mestranol similarity, amlodipine MPO, perindopril MPO, deco hop, median1, and isomers c9h10n2o2pf2cl) under 2500 oracle calls.

## 5.2 Experimental Results

To evaluate the overall performance of our method, we examine various metrics across multiple oracles. Table 2 presents a comparative analysis based on Average Top 10, AUC Top 1, AUC Top 10, and AUC Top 100 scores. We observe that for most oracles, Gradient GA outperforms Graph GA, MIMOSA, MARS, and SMILES GA. Our method consistently achieves either the best or second-best performance. Specifically, in Average Top 10, Gradient GA demonstrates the highest performance, with Graph GA following behind. This highlights the advantage of incorporating gradient information, enabling more efficient and effective exploration of the local search space. The superiority of Gradient GA is further supported by the AUC Top K scores, which reward methods that reach high values with fewer oracle calls. For AUC Top 1 and AUC Top 10, Gradient GA dominates the rankings, demonstrating faster convergence and better optimization. In AUC Top 100, Gradient GA achieves a top score, tying with SMILES GA. However, while SMILES GA performs well in this metric, it significantly underperforms in Average scores. These results confirm that Gradient GA is the most effective method overall for optimizing molecular properties across different oracles.

### 5.2.1 Oracle Call Efficiency

To demonstrate that using gradient information accelerates convergence, we conduct experiments measuring AUC Top 10 and AUC Top 100 scores as the number of oracle calls increases. All methods are evaluated with 2,500 oracle calls over 5 runs. Our primary focus is on the bio-objective Mestranol Similarity. Figures 3 and 4 show that after the initialization phase, where each method achieves a baseline score based on the initial oracle calls, Gradient GA consistently outperforms almost all other methods at each step. This indicates that Gradient GA is not only more effective at finding optimal values but also more efficient, due to its use of gradient guidance rather than random walk exploration.

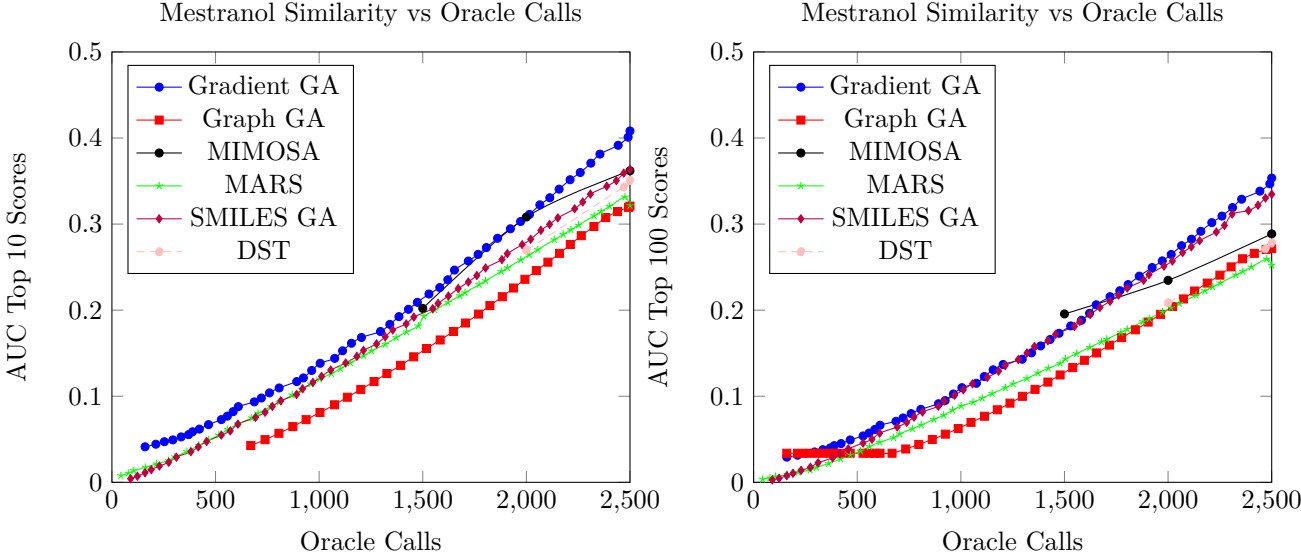

Figure 3: Comparison of Mestranol similarity AUC Top 10 scores as the number of oracle calls increases.

Figure 4: Comparison of Mestranol similarity AUC Top 100 scores as the number of oracle calls increases.

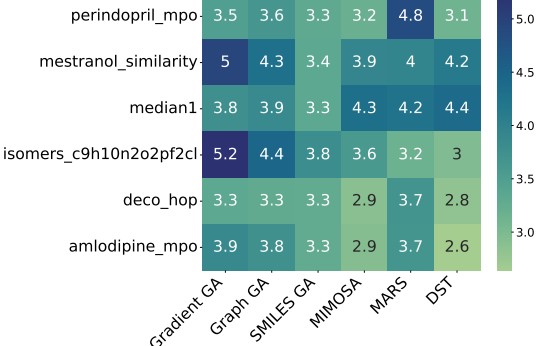

Figure 5: Heatmap of synthetic accessibility (SA) score for all methods and oracles (lower is better).

Figure 6: Heatmap of diversity score for all methods and oracles (higher is better).

### 5.2.2 Synthetic Accessibility and Diversity

We further analyze additional metrics, synthetic accessibility (SA) in Figure 5 and diversity in Figure 6. It is important to note that these metrics are *not* explicitly optimized in our objective function. Therefore, their performance is a byproduct of the discovered molecules rather than a direct outcome of our method. From Figure 5, we observe that the SA scores of Gradient GA are comparable to those of Graph GA, indicating that there is no significant trade-off between improved performance and SA score. Additionally, in terms of overall SA performance, Gradient GA is also close to DST, another gradient-based method. In Figure 6, we observe that the diversity score for Gradient GA is lower than that of other methods. This outcome is expected, as our approach samples molecules near high-performing parent molecules. While this may reduce diversity, it can be advantageous when the goal is to perform a fine-grained local search over good regions. Compared to other variations on genetic algorithms (GA), the diversity for performance tradeoff of Gradient GA is more moderate. For example, there are two methods, Genetic GFN (Kim et al., 2024) which leverages GFlowNets to improve molecule selection within GA and MOL GA (Tripp & Hernández-Lobato, 2023), a

method that introduces an optimized version of standard GA aimed at achieving higher AVG and AUC scores on the PMO benchmark. They both perform better than Gradient GA, but at higher amounts of oracle calls, they perform similarly, but their diversity is much worse at all amounts of calls. Detailed results are in Appendix F.

### 5.2.3 Molecules Generated by Gradient GA

We have included the Top 10 molecules generated by Gradient GA for the mestranol similarity objective in Figure 7. We can observe some structural similarity between the Top-10 molecules as they have been sampled from the same run. Such observation implies that one compatible set of parents can produce a high proportion of good molecules. We further extend the study of good molecules for different methods in Appendix B.

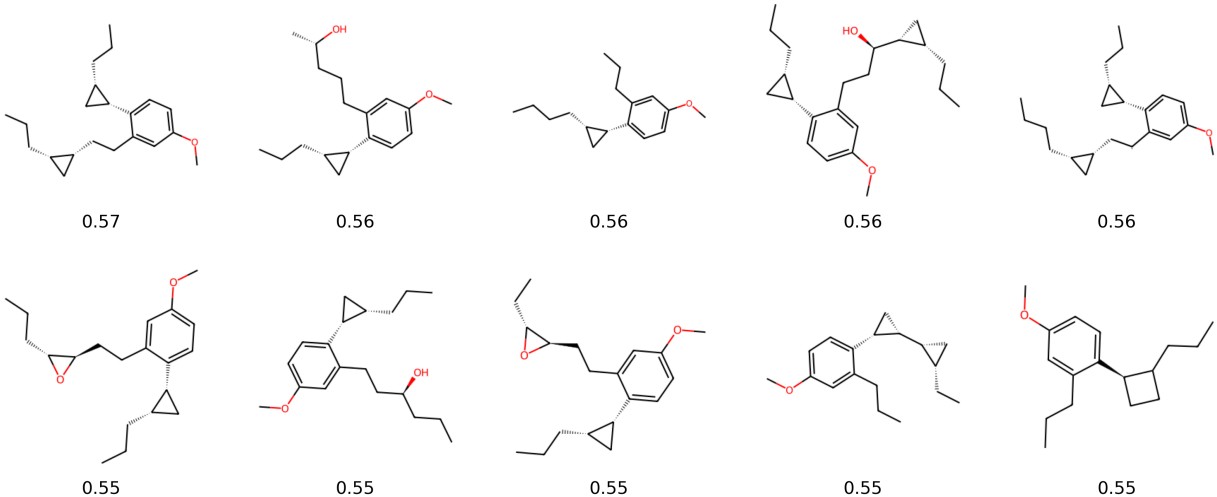

Figure 7: Top 10 molecules generated by Gradient GA for the bio-activity objective mestranol similarity with their associated score underneath each molecule.

### 5.2.4 Computational Cost Comparison

Gradient computation in Gradient GAintroduces additional overhead compared to standard GA. However, the cost remains modest and practical: for a 1000-call run, GA required about 1 minute, while our method required 2-3 minutes. In contrast, RL-based methods required 4-5 minutes, making our approach about $0.6\times$ their runtime. Thus, while slightly more expensive than GA, our method remains significantly more efficient than RL-based baselines while achieving superior diversity and optimization tradeoffs.

## 6 Conclusion and Future Work

Genetic Algorithm (GA) is one of the most widely used approaches in drug molecular design, thanks to its flexibility in navigating the molecular space. However, GA usually suffers from slow and unstable convergence due to its undirected random exploration. We address this problem by introducing a novel approach called Gradient GA, in which each proposed sample iteratively progresses toward the optimal solution. Our method leverages Discrete Langevin Proposal (DLP) as the base sampler, enabling gradient-based exploration in the discrete molecular space. Extensive experimental results confirm that our proposed approach achieves faster and superior convergence compared to state-of-the-art methods.

Future work can expand the current method in the following aspects: (1) explore the effect of utilizing gradient information for generated molecule populations; (2) explore better ways to fit both parents into DLP; (3) explore DLP-oriented molecular optimizations that incorporate the Metropolis-Hastings criterion; (4) explore multi-objective optimization (preliminary results are given in Appendix G).

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

# A  Additional Experimental Results

## A.1  Storing molecules for retraining

The tables below are results from the same experimental setup described in Table 2. The higher the score, the better for all metrics except Average SA.

For most oracles, Gradient GA achieves the highest performance in Average Top 1, while for Average Top 100 and AUC Top 1, it consistently outperforms all other methods across all oracles. Graph GA and DST continue to perform well overall, ranking near the top in almost every oracle. MIMOSA and MARS follow closely, occasionally achieving top-tier results. On the other hand, SMILES GA performs the worst in terms of optimization metrics but ranks near the top for diversity and achieves the lowest SA score.

Table 3: mestranol similarity Results

| Method | Average Top 1 | Top 10 | Top 100 | AUC Top 1 | Top 10 | Top 100 | Average SA | Diversity |
|---|---|---|---|---|---|---|---|---|
| Gradient GA | **0.5367±0.0421** | **0.5130±0.0393** | **0.4673±0.0293** | **0.4433±0.0310** | **0.4082±0.0315** | **0.3534±0.0355** | 4.9647±0.3605 | 0.6692±0.0294 |
| Graph GA | 0.4726±0.0241 | 0.4452±0.0241 | 0.4041±0.0237 | 0.3556±0.0268 | 0.3208±0.0199 | 0.2717±0.0147 | 4.3011±0.5876 | 0.7660±0.0380 |
| SMILES GA | 0.2913±0.0178 | 0.2582±0.0097 | 0.1703±0.0074 | 0.3777±0.0381 | 0.3634±0.0352 | 0.3347±0.0279 | **3.3519±0.0524** | **0.8711±0.0065** |
| MIMOSA | 0.4784±0.0456 | 0.4262±0.0246 | 0.3596±0.0371 | 0.4162±0.0115 | 0.3619±0.0181 | 0.2887±0.0252 | 3.9385±0.1149 | 0.7782±0.0721 |
| MARS | 0.3837±0.0004 | 0.3411±0.0160 | 0.2803±0.0205 | 0.3760±0.0003 | 0.3215±0.0096 | 0.2523±0.0081 | 3.9512±0.2232 | 0.8594±0.0032 |
| DST | 0.4591±0.0305 | 0.4131±0.0179 | 0.3654±0.0126 | 0.4148±0.0323 | 0.3507±0.0088 | 0.2780±0.0029 | 4.1479±0.0984 | 0.8198±0.0079 |

Table 4: median1 Results

| Method | Average Top 1 | Top 10 | Top 100 | AUC Top 1 | Top 10 | Top 100 | Average SA | Diversity |
|---|---|---|---|---|---|---|---|---|
| Gradient GA | **0.3185±0.0113** | **0.3033±0.0074** | **0.2688±0.0086** | **0.2581±0.0115** | **0.2298±0.0151** | **0.1906±0.0183** | 3.7551±0.2584 | 0.6974±0.0368 |
| Graph GA | 0.2779±0.0251 | 0.2599±0.0182 | 0.2292±0.0127 | 0.2315±0.0206 | 0.1959±0.0148 | 0.1442±0.0070 | 3.8609±0.1770 | 0.7462±0.0425 |
| SMILES GA | 0.1587±0.0370 | 0.1310±0.0172 | 0.0674±0.0067 | 0.1832±0.0281 | 0.1795±0.0272 | 0.1697±0.0251 | **3.3444±0.1014** | **0.8700±0.0074** |
| MIMOSA | 0.2802±0.0126 | 0.2391±0.0080 | 0.1948±0.0091 | 0.2271±0.0103 | 0.1969±0.0044 | 0.1537±0.0030 | 4.3154±0.1356 | 0.8148±0.0209 |
| MARS | 0.2322±0.0201 | 0.2094±0.0181 | 0.1777±0.0234 | 0.2239±0.0140 | 0.2019±0.0116 | 0.1671±0.0158 | 4.2508±0.3090 | 0.8458±0.0196 |
| DST | 0.2512±0.0376 | 0.2179±0.0162 | 0.1734±0.0049 | 0.2097±0.0086 | 0.1765±0.0021 | 0.1331±0.0024 | 4.4038±0.2342 | 0.8554±0.0098 |

Table 5: amlodipine MPO Results

| Method | Average Top 1 | Top 10 | Top 100 | AUC Top 1 | Top 10 | Top 100 | Average SA | Diversity |
|---|---|---|---|---|---|---|---|---|
| Gradient GA | **0.5880±0.0397** | **0.5667±0.0336** | **0.5400±0.0296** | **0.5614±0.0177** | **0.5176±0.0187** | 0.4658±0.0199 | 3.8677±0.3122 | 0.6176±0.0478 |
| Graph GA | 0.5772±0.0409 | 0.5605±0.0364 | 0.5232±0.0339 | 0.5067±0.0270 | 0.4734±0.0215 | 0.4152±0.0142 | 3.7496±0.1504 | 0.7571±0.0421 |
| SMILES GA | 0.4782±0.0198 | 0.4480±0.0161 | 0.2242±0.0370 | 0.5016±0.0156 | 0.4956±0.0143 | **0.4748±0.0158** | 3.2925±0.0951 | **0.8711±0.0064** |
| MIMOSA | 0.5633±0.0222 | 0.5245±0.0143 | 0.4996±0.0132 | 0.5431±0.0261 | 0.4953±0.0109 | 0.4436±0.0075 | 2.9351±0.2224 | 0.7430±0.0310 |
| MARS | 0.5079±0.0301 | 0.4843±0.0210 | 0.4412±0.0295 | 0.4812±0.0144 | 0.4583±0.0098 | 0.3816±0.0157 | 3.6660±0.3345 | 0.8504±0.0169 |
| DST | 0.5609±0.0285 | 0.5192±0.0122 | 0.4704±0.0069 | 0.5411±0.0303 | 0.4908±0.0115 | 0.4257±0.0044 | **2.6388±0.0770** | 0.8349±0.0050 |

Table 6: perindopril MPO Results

| Method | Average Top 1 | Top 10 | Top 100 | AUC Top 1 | Top 10 | Top 100 | Average SA | Diversity |
|---|---|---|---|---|---|---|---|---|
| Gradient GA | 0.4856±0.0273 | 0.4786±0.0257 | 0.4620±0.0287 | 0.4542±0.0164 | 0.4361±0.0176 | 0.3882±0.0193 | 3.4861±0.4956 | 0.5720±0.0691 |
| Graph GA | 0.4985±0.0174 | 0.4788±0.0067 | 0.4438±0.0049 | 0.4519±0.0055 | 0.4317±0.0045 | 0.3770±0.0049 | 3.6484±0.1822 | 0.7766±0.0157 |
| SMILES GA | 0.4228±0.0090 | 0.3698±0.0117 | 0.1870±0.0185 | 0.4346±0.0124 | 0.4271±0.0115 | 0.4065±0.0102 | 3.3328±0.0955 | 0.8694±0.0080 |
| MIMOSA | 0.4719±0.0166 | 0.4629±0.0176 | 0.4366±0.0169 | 0.4500±0.0144 | 0.4289±0.0116 | 0.3783±0.0085 | 3.1907±0.2006 | 0.7453±0.0284 |
| MARS | 0.4738±0.0158 | 0.4564±0.0167 | 0.4250±0.0147 | 0.4538±0.0087 | 0.4278±0.0065 | 0.3648±0.0042 | 4.8547±0.0727 | 0.8272±0.0078 |
| DST | 0.4774±0.0101 | 0.4615±0.0100 | 0.4197±0.0177 | 0.4530±0.0041 | 0.4210±0.0041 | 0.3564±0.0028 | 3.1428±0.1120 | 0.7997±0.0294 |

Table 7: deco hop Results

| Method | Average Top 1 | Top 10 | Top 100 | AUC Top 1 | Top 10 | Top 100 | Average SA | Diversity |
|---|---|---|---|---|---|---|---|---|
| Gradient GA | 0.6116±0.0013 | 0.6026±0.0053 | **0.5945±0.0068** | **0.5883±0.0032** | 0.5763±0.0050 | **0.5602±0.0053** | 3.3425±0.2122 | 0.6444±0.0420 |
| Graph GA | 0.6101±0.0039 | **0.6039±0.0043** | 0.5923±0.0027 | 0.5186±0.0037 | 0.5028±0.0032 | 0.4708±0.0033 | 3.3022±0.0970 | 0.7870±0.0220 |
| SMILES GA | 0.5733±0.0062 | 0.5548±0.0059 | 0.5178±0.0082 | 0.5862±0.0047 | **0.5817±0.0042** | 0.5733±0.0036 | 3.2847±0.1118 | **0.8699±0.0082** |
| MIMOSA | 0.6071±0.0072 | 0.6008±0.0053 | 0.5906±0.0046 | 0.5882±0.0061 | 0.5773±0.0035 | 0.5600±0.0021 | 2.8906±0.2936 | 0.7428±0.0481 |
| MARS | 0.6014±0.0069 | 0.5944±0.0070 | 0.5830±0.0095 | 0.5830±0.0227 | 0.5711±0.0301 | 0.5493±0.0421 | 3.7003±0.1504 | 0.8182±0.0646 |
| DST | **0.6128±0.0118** | 0.6034±0.0083 | 0.5878±0.0043 | 0.5860±0.0071 | 0.5721±0.0025 | 0.5518±0.0009 | **2.8406±0.1211** | 0.8305±0.0060 |

Table 8: isomers c9h10n2o2pf2cl Results

| Method | Average Top 1 | Top 10 | Top 100 | AUC Top 1 | Top 10 | Top 100 | Average SA | Diversity |
|--------|---------------|--------|---------|-----------|--------|---------|------------|-----------|
| Gradient GA | **0.8258±0.0940** | 0.7783±0.0959 | 0.6697±0.0993 | 0.6628±0.0731 | 0.5444±0.0693 | 0.4033±0.0614 | 5.2421±0.3124 | 0.7908±0.0172 |
| Graph GA | 0.7593±0.1157 | 0.7222±0.1119 | 0.6438±0.0950 | 0.6648±0.0957 | 0.5436±0.0770 | 0.3891±0.0425 | 4.4437±0.3496 | 0.7933±0.0676 |
| SMILES GA | 0.5597±0.2924 | 0.3180±0.3583 | 0.1972±0.4036 | **0.8244±0.0848** | **0.7825±0.0752** | **0.7055±0.0668** | 3.7996±1.1599 | 0.8320±0.0890 |
| MIMOSA | 0.8146±0.0818 | **0.7866±0.0824** | **0.6848±0.0541** | 0.6965±0.0562 | 0.5949±0.0440 | 0.3965±0.0265 | 3.6001±0.2458 | 0.8231±0.0300 |
| MARS | 0.7268±0.1260 | 0.6639±0.1606 | 0.5268±0.2389 | 0.6751±0.1032 | 0.5989±0.1057 | 0.4424±0.1499 | 3.1615±0.8526 | 0.8335±0.0970 |
| DST | 0.7703±0.0482 | 0.6748±0.0304 | 0.5028±0.0329 | 0.6305±0.0435 | 0.4932±0.0216 | 0.2293±0.0093 | **2.9831±0.1486** | **0.8735±0.0033** |

## A.2 Retrained only on newly generated molecules between retraining phases

The overall experimental setup includes 10,000 oracle calls, 5 runs, and early stopping enabled. The parameters for Gradient GA are set to 200 epochs, with $D''$ being cleared after each retraining. We have run REINVENT (Olivecrona et al., 2017), which is a reinforcement learning method, instead of DST. Additionally, we evaluate another metric, Diversity, which measures the Tanimoto Similarity between two molecules. The higher the score, the better for all metrics except Average SA. This setup overall performs worse than when $D''$ is saved; however, it is slightly faster in runtime, due to having fewer molecules for retraining.

From the results, we can see that Gradient GA and Graph GA are near the top in terms of performance with REINVENT. These results follow the results gathered from PMO (Gao et al., 2022b). We notice that Gradient GA usually performs better than Graph GA whenever it is dealing with a molecular objective that is on the lower end, so the exploration near optimal molecules matters more than a random walk behavior may have a harder time finding the optimal molecules. Overall, the best-performing models are REINVENT, Gradient GA, and Graph GA.

Table 9: perindopril mpo Results

| Method | Average Top 1 | Top 10 | Top 100 | AUC Top 1 | Top 10 | Top 100 | Average SA | Diversity |
|--------|---------------|--------|---------|-----------|--------|---------|------------|-----------|
| Gradient GA | 0.5741±0.0104 | 0.5613±0.0109 | 0.5443±0.0113 | **0.5228±0.0113** | **0.5102±0.0109** | **0.4874±0.0111** | 4.7354±0.4388 | 0.5477±0.0594 |
| Graph GA | 0.5350±0.0612 | 0.5245±0.0585 | 0.5000±0.0528 | 0.5010±0.0393 | 0.4848±0.0371 | 0.4515±0.0339 | 3.9596±0.1366 | 0.6685±0.1190 |
| SMILES GA | 0.4495±0.0144 | 0.4495±0.0144 | 0.4478±0.0145 | 0.4455±0.0127 | 0.4433±0.0123 | 0.4355±0.0121 | 4.7639±0.4927 | 0.4968±0.0538 |
| MARS | 0.4793±0.0137 | 0.4647±0.0128 | 0.4357±0.0135 | 0.4751±0.0112 | 0.4570±0.0098 | 0.4173±0.0090 | 5.0202±0.2178 | **0.8231±0.0053** |
| MIMOSA | 0.4703±0.0177 | 0.4569±0.0107 | 0.4403±0.0076 | 0.3247±0.0046 | 0.3116±0.0042 | 0.2856±0.0031 | **3.9169±0.2835** | 0.6964±0.0230 |
| REINVENT | **0.6196±0.0460** | **0.6164±0.0485** | **0.6132±0.0511** | 0.4563±0.0764 | 0.4428±0.0755 | 0.4188±0.0746 | 4.1524±0.3994 | 0.3490±0.0603 |

Table 10: mestranol similarity Results

| Method | Average Top 1 | Top 10 | Top 100 | AUC Top 1 | Top 10 | Top 100 | Average SA | Diversity |
|--------|---------------|--------|---------|-----------|--------|---------|------------|-----------|
| Gradient GA | 0.7348±0.0321 | 0.7140±0.0282 | 0.6618±0.0298 | **0.6179±0.0401** | **0.5833±0.0349** | **0.5273±0.0318** | 4.6019±0.3707 | 0.5329±0.0394 |
| Graph GA | 0.7335±0.1325 | 0.6926±0.1161 | 0.6401±0.0928 | 0.6123±0.0516 | 0.5727±0.0429 | 0.5173±0.0307 | 4.2850±0.5503 | 0.5599±0.0471 |
| SMILES GA | 0.4488±0.0456 | 0.4477±0.0454 | 0.4441±0.0433 | 0.4196±0.0425 | 0.4129±0.0419 | 0.4007±0.0395 | 5.1324±0.6792 | 0.5278±0.1021 |
| MARS | 0.4142±0.0662 | 0.3782±0.0718 | 0.3202±0.0756 | 0.4059±0.0638 | 0.3671±0.0655 | 0.3047±0.0645 | 4.0433±0.3763 | **0.8546±0.0105** |
| MIMOSA | 0.5239±0.0145 | 0.4907±0.0147 | 0.4450±0.0186 | 0.5079±0.0010 | 0.4688±0.0021 | 0.4069±0.0044 | 3.8981±0.3227 | 0.8052±0.0398 |
| REINVENT | **0.7838±0.0823** | **0.7809±0.0835** | **0.7627±0.0834** | 0.3924±0.0836 | 0.3709±0.0838 | 0.3346±0.0824 | **3.6929±0.4919** | 0.2989±0.0471 |

Table 11: median1 Results

| Method | Average Top 1 | Top 10 | Top 100 | AUC Top 1 | Top 10 | Top 100 | Average SA | Diversity |
|--------|---------------|--------|---------|-----------|--------|---------|------------|-----------|
| Gradient GA | 0.3923±0.0173 | 0.3714±0.0183 | 0.3378±0.0095 | **0.3376±0.0215** | **0.3108±0.0153** | **0.2789±0.0125** | 4.0045±0.1275 | 0.6191±0.0278 |
| Graph GA | 0.3093±0.0345 | 0.2906±0.0257 | 0.2593±0.0180 | 0.2922±0.0311 | 0.2680±0.0228 | 0.2318±0.0158 | 4.0053±0.1491 | 0.7091±0.0363 |
| SMILES GA | 0.2004±0.0300 | 0.1989±0.0298 | 0.1980±0.0293 | 0.1977±0.0296 | 0.1936±0.0284 | 0.1889±0.0268 | 6.0024±1.0691 | 0.6396±0.0382 |
| MARS | 0.2322±0.0201 | 0.2094±0.0181 | 0.1777±0.0234 | 0.2239±0.0140 | 0.2019±0.0116 | 0.1671±0.0158 | 4.2508±0.3090 | **0.8458±0.0196** |
| MIMOSA | 0.3275±0.0130 | 0.3011±0.0036 | 0.2686±0.0003 | 0.2675±0.0043 | 0.2278±0.0013 | 0.1654±0.0013 | **3.9701±0.0735** | 0.7643±0.0091 |
| REINVENT | **0.4579±0.0004** | **0.4384±0.0193** | **0.4181±0.0344** | 0.2571±0.0514 | 0.2282±0.0447 | 0.1852±0.0373 | 4.7140±0.6151 | 0.3136±0.1278 |

Table 12: isomers c9h10n2o2pf2cl Results

| Method | Average Top 1 | Top 10 | Top 100 | AUC Top 1 | Top 10 | Top 100 | Average SA | Diversity |
|---|---|---|---|---|---|---|---|---|
| Gradient GA | **0.9394±0.0000** | 0.9237±0.0196 | 0.8733±0.0264 | 0.8286±0.0704 | 0.7878±0.0758 | 0.7018±0.0808 | 5.4939±0.2025 | 0.7481±0.0389 |
| Graph GA | 0.9137±0.0291 | 0.8869±0.0285 | 0.8316±0.0177 | 0.8441±0.0206 | 0.8024±0.0158 | 0.7171±0.0141 | 4.5273±0.3671 | 0.7995±0.0265 |
| SMILES GA | 0.9341±0.0366 | **0.9310±0.0363** | **0.8895±0.0457** | **0.8955±0.0351** | **0.8640±0.0352** | **0.8083±0.0450** | 5.8310±0.2092 | 0.7161±0.0519 |
| MARS | 0.7268±0.1260 | 0.6639±0.1606 | 0.5268±0.2389 | 0.6751±0.1032 | 0.5989±0.1057 | 0.4424±0.1499 | **3.1615±0.8526** | **0.8335±0.0970** |
| MIMOSA | 0.8352±0.0458 | 0.8081±0.0637 | 0.7564±0.0661 | 0.6092±0.0248 | 0.5745±0.0244 | 0.5007±0.0200 | 4.1175±0.8821 | 0.7669±0.0722 |
| REINVENT | 0.9166±0.0312 | 0.9030±0.0267 | 0.8718±0.0256 | 0.3557±0.0381 | 0.3331±0.0367 | 0.2874±0.0323 | 3.1677±0.7806 | 0.6673±0.0776 |

Table 13: deco hop Results

| Method | Average Top 1 | Top 10 | Top 100 | AUC Top 1 | Top 10 | Top 100 | Average SA | Diversity |
|---|---|---|---|---|---|---|---|---|
| Gradient GA | 0.6475±0.0092 | 0.6413±0.0091 | 0.6353±0.0092 | 0.6200±0.0060 | 0.6124±0.0059 | 0.6024±0.0060 | 3.9402±0.4094 | 0.5562±0.0621 |
| Graph GA | 0.6794±0.1400 | 0.6738±0.1420 | 0.6632±0.1411 | 0.6452±0.0840 | 0.6326±0.0799 | 0.6120±0.0729 | 3.2752±0.2231 | 0.7074±0.1102 |
| SMILES GA | 0.6147±0.0061 | 0.6147±0.0061 | 0.6144±0.0059 | 0.5927±0.0060 | 0.5903±0.0060 | 0.5844±0.0057 | 4.9016±0.5634 | 0.5258±0.0800 |
| MARS | 0.6014±0.0069 | 0.5944±0.0070 | 0.5830±0.0095 | 0.5830±0.0227 | 0.5711±0.0301 | 0.5493±0.0421 | 3.7003±0.1504 | **0.8182±0.0646** |
| MIMOSA | 0.6051±0.0122 | 0.6032±0.0130 | 0.5896±0.0100 | 0.5428±0.0053 | 0.5173±0.0049 | 0.4726±0.0029 | 4.1345±0.2664 | 0.6712±0.0811 |
| REINVENT | **0.8014±0.1476** | **0.7915±0.1430** | **0.7853±0.1413** | **0.6577±0.0628** | **0.6415±0.0543** | **0.6231±0.0478** | **3.0143±0.2320** | 0.4356±0.0308 |

Table 14: amlodipine mpo Results

| Method | Average Top 1 | Top 10 | Top 100 | AUC Top 1 | Top 10 | Top 100 | Average SA | Diversity |
|---|---|---|---|---|---|---|---|---|
| Gradient GA | 0.6942±0.0499 | 0.6848±0.0510 | 0.6698±0.0456 | 0.6309±0.0361 | 0.6120±0.0348 | 0.5845±0.0309 | 4.0448±0.0917 | 0.4402±0.0859 |
| Graph GA | 0.7150±0.0407 | 0.6958±0.0290 | 0.6730±0.0258 | **0.6569±0.0202** | **0.6350±0.0174** | **0.6004±0.0131** | 3.8835±0.1383 | 0.5982±0.0422 |
| SMILES GA | 0.5254±0.0335 | 0.5235±0.0375 | 0.5216±0.0379 | 0.5130±0.0295 | 0.5079±0.0324 | 0.4995±0.0317 | 4.6961±0.2230 | 0.5945±0.0615 |
| MARS | 0.5081±0.0303 | 0.4902±0.0279 | 0.4488±0.0392 | 0.5014±0.0259 | 0.4818±0.0223 | 0.4311±0.0315 | 3.6973±0.3809 | **0.8430±0.0302** |
| MIMOSA | 0.6045±0.0118 | 0.5789±0.0166 | 0.5540±0.0134 | 0.5908±0.0211 | 0.5429±0.0174 | 0.4979±0.0084 | 4.0689±0.5939 | 0.6628±0.0650 |
| REINVENT | **0.7382±0.0453** | **0.7334±0.0431** | **0.7259±0.0400** | 0.5576±0.0574 | 0.5409±0.0553 | 0.5147±0.0527 | **3.2323±0.2277** | 0.3946±0.0670 |

## B   Molecules Generated by Various Methods

Figures 8 - 11 show the top 10 molecules generated for the mestranol similarity by Gradient GA, SMILES GA, Graph GA, MARS, and MIMOSA. The molecules proposed by Gradient GA, SMILES GA, and Graph GA show a strong structural resemblance. Although MARS and MIMOSA-proposed molecules bear more uniqueness, they yield a lower score for the product metric, denoting a poor choice of molecules. From the figures, it is evident that Gradient GA is exploring similar molecules to Graph GA, while further exploring an area around its top-performing molecules.

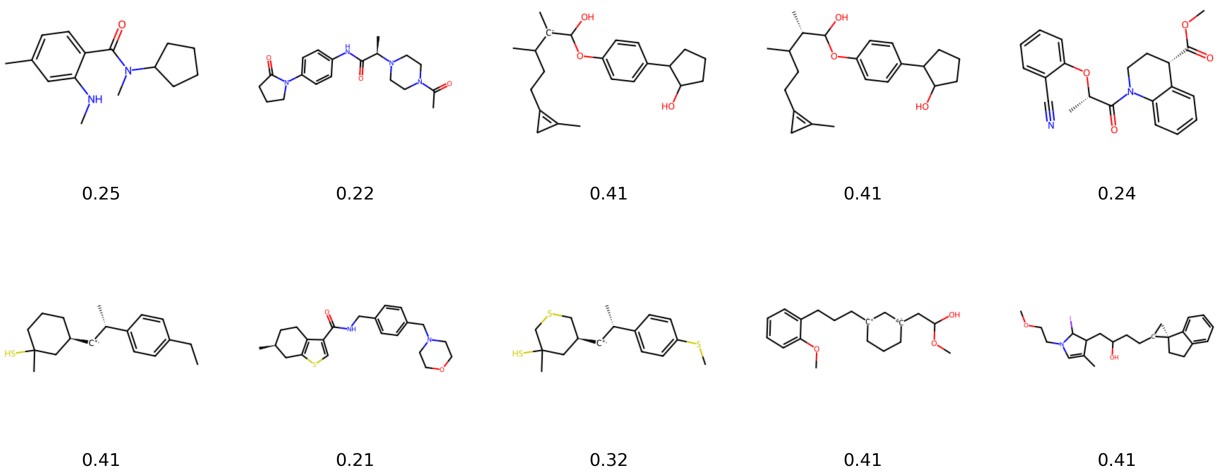

Figure 8: Top 10 molecules generated by SMILES GA for the bio-activity objective mestranol similarity with their associated score underneath each molecule.

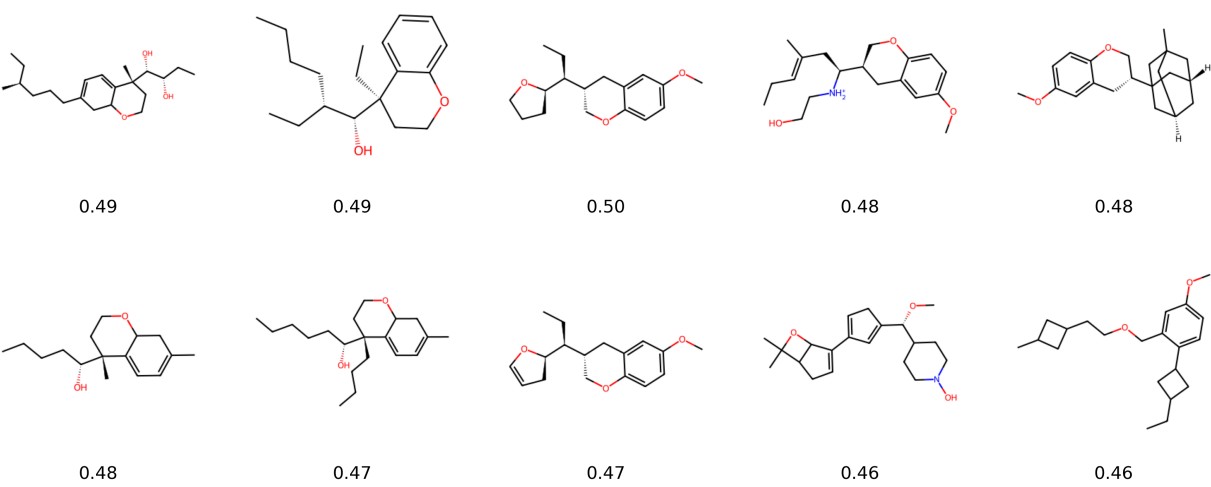

Figure 9: Top 10 molecules generated by Graph GA for the bio-activity objective mestranol similarity with their associated score underneath each molecule.

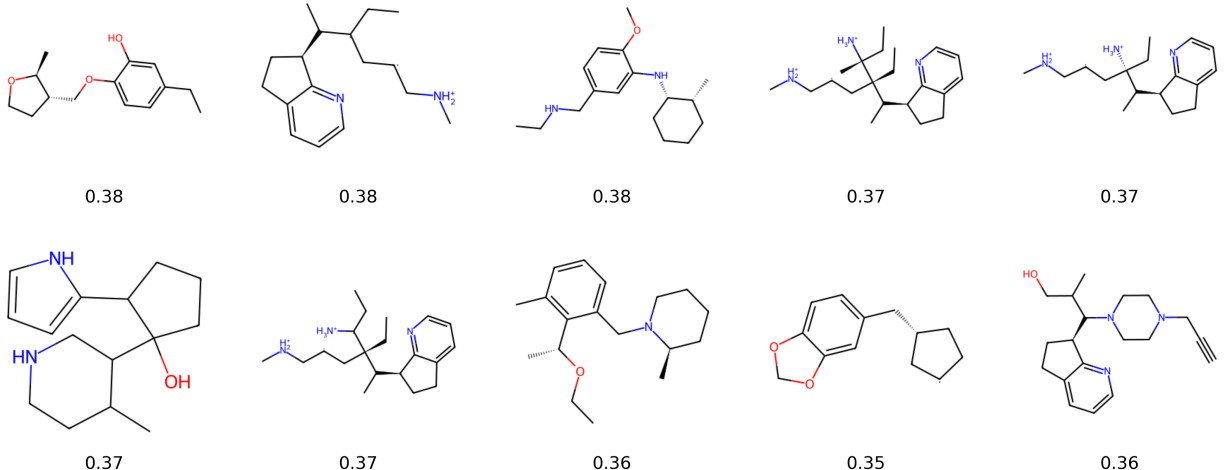

Figure 10: Top 10 molecules generated by MARS for the bio-activity objective mestranol similarity with their associated score underneath each molecule.

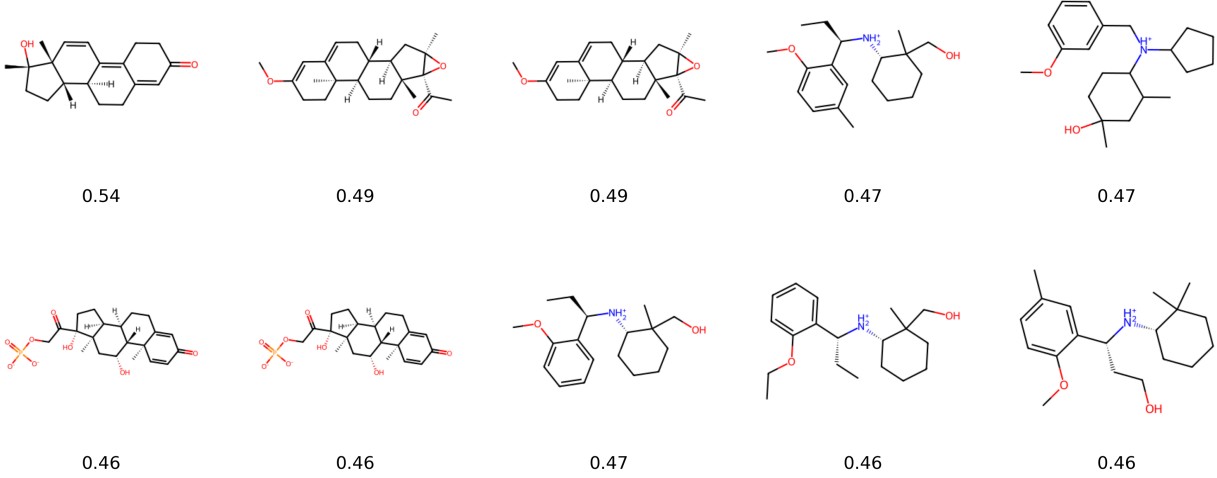

Figure 11: Top 10 molecules generated by MIMOSA for the bio-activity objective mestranol similarity with their associated score underneath each molecule.

## C   Comparison of Including $\mathcal{O}(X)$ in Gradient

In the table below, we show the results of using $\nabla U(v) = \nabla f(v)$ vs $\nabla U(v) = \frac{\nabla f(v)}{\mathcal{O}(X)}$. We notice from the results, using $\mathcal{O}(x)$ leads to a better score throughout all metrics.

Table 15: Comparison of Average Top 10, AUC Top 1, AUC Top 10, and AUC Top 100 with GuacaMol objective, mestranol similarity, under 2500 oracle calls. The best Gradient GA setup is **bolded**. We conduct five independent runs using different random seeds for both versions of Gradient GA, and report the average scores and their standard deviation.

| Gradient GA | mestranol similarity | | | |
|---|---|---|---|---|
| | Average Top 10 | AUC Top 1 | AUC Top 10 | AUC Top 100 |
| With $\mathcal{O}(x)$ | **0.5130±0.0393** | **0.4433±0.0310** | **0.4082±0.0315** | **0.3534±0.0355** |
| Without $\mathcal{O}(x)$ | 0.5064±0.0312 | 0.4433±0.0319 | 0.4072±0.0367 | 0.3501±0.0419 |

## D   Surrogate $\mathcal{M}$ Error Analysis

In this section, we discuss the effects of surrogate loss introduced by $\mathcal{M}$. We consider two different sets for validation: (i) Validation Training Set: usual validation set, kept from training samples of $\mathcal{M}$, (ii) Validation Known Samples: validation set constructed from the current population. The size of the validation set is 10% of the respective sets. We observe the training loss, both validation losses with respect to oracle calls in Figure 12. While Validation Training Set Loss shows a similar trend to Training Set Loss, Validation Known Samples Loss decreases steadily over time. This shows that the approximation from $\mathcal{M}$ is relatively accurate even in more unknown situations for the neural network, demonstrating good estimation capability.

Figure 12: Comparison of the total training loss, total validation loss from the training set, and also the validation loss from all known samples with 10000 oracle call runs.

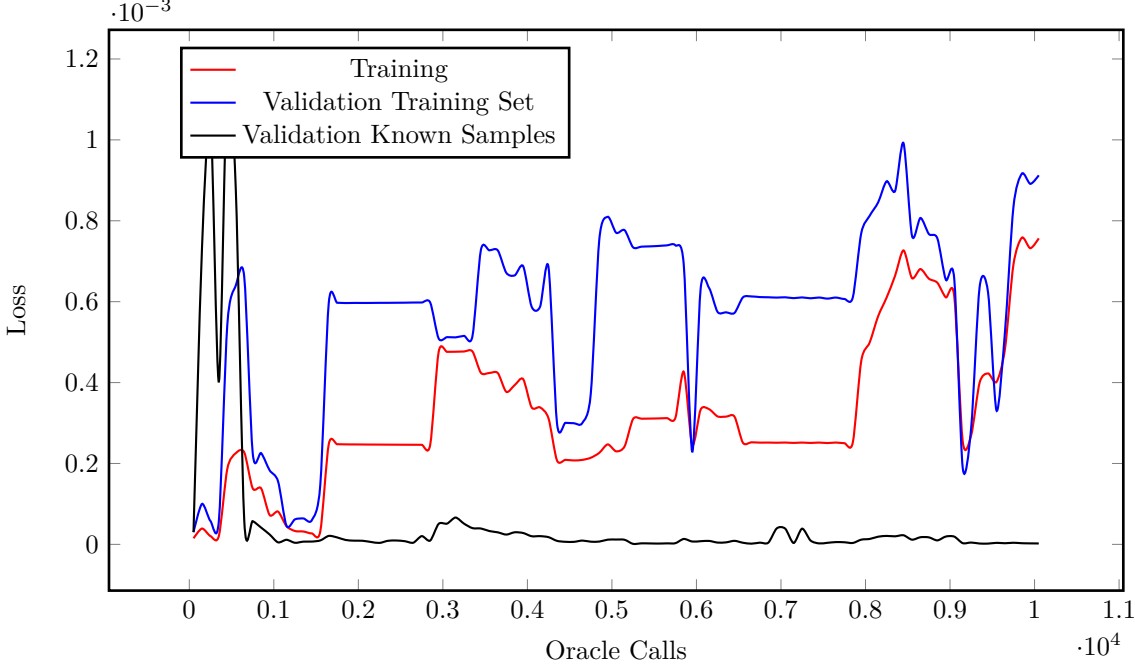

# E Comparison between Parent Selection Methods

In the table below, we show the results of using the best parent and averaging the parents for the gradient computation. We notice from the results, using the best parent leads to a better score throughout all metrics except diversity.

Table 16: Comparison of Average Top 10, AUC Top 1, AUC Top 10, AUC Top 100, and Diversity with GuacaMol objective, Amlodipine MPO, under 1000 oracle calls. The best Gradient GA setup is **bolded**. We conduct five independent runs using different random seeds for both versions of Gradient GA, and report the average scores and their standard deviation.

| Gradient GA | Amlodipine MPO | | | | |
|---|---|---|---|---|---|
| | Average Top 1 | AUC Top 1 | AUC Top 10 | AUC Top 100 | Diversity |
| Best Parent | **0.5645±0.0071** | **0.5294±0.0063** | **0.4681±0.0165** | **0.3929±0.0149** | 0.6855±0.101 |
| Averaging Parents | 0.5337±0.0174 | 0.4979±0.0198 | 0.4616±0.0057 | 0.3908±0.008 | **0.7425±0.0257** |

# F Variations of Graph GA Algorithms

In the tables below, we show Gradient GA compared to other recent GA methods under the same experimental setup as in Table 2. From the results, it is clear that our method achieves higher diversity compared to Genetic GFN and MOL GA, though it lags behind in terms of AVG and AUC scores. A notable trend emerges when comparing performance at 2,500 versus 10,000 oracle calls: the gap in AVG and AUC scores between Gradient GA and other GA-based methods narrows significantly at 10,000 calls. While all methods experience a drop in diversity as the number of oracle calls increases, Gradient GA consistently maintains a higher diversity at both checkpoints.

We believe that Gradient GA offers a balanced trade-off, achieving competitive AVG and AUC scores while preserving diversity, especially at higher oracle call budgets. This is particularly important given that, as shown in the previous section, GA-based methods generally exhibit lower diversity than state-of-the-art non-GA approaches. Maintaining molecular diversity is crucial for practical applications, as it ensures the recommended molecules are not overly similar in structure.

| Metric | Method | Top 1 | Top 10 | Top 100 | AUC Top 1 | AUC Top 10 | Diversity Top 100 |
|---|---|---|---|---|---|---|---|
| Mestranol Similarity | Genetic GFN | 0.668 ± 0.024 | 0.668 ± 0.024 | 0.650 ± 0.044 | **0.614 ± 0.053** | 0.545 ± 0.021 | 0.458 ± 0.027 |
| | MOL GA | **0.728 ± 0.040** | **0.687 ± 0.051** | **0.622 ± 0.063** | 0.572 ± 0.058 | **0.538 ± 0.053** | 0.571 ± 0.113 |
| | Gradient GA | 0.537 ± 0.042 | 0.513 ± 0.039 | 0.467 ± 0.029 | 0.443 ± 0.031 | 0.408 ± 0.032 | **0.669 ± 0.029** |
| Amlodipine MPO | Genetic GFN | **0.671 ± 0.044** | 0.660 ± 0.050 | 0.635 ± 0.055 | 0.603 ± 0.039 | **0.584 ± 0.039** | 0.530 ± 0.135 |
| | MOL GA | 0.656 ± 0.041 | **0.663 ± 0.036** | **0.639 ± 0.033** | **0.605 ± 0.038** | 0.592 ± 0.036 | 0.492 ± 0.015 |
| | Gradient GA | 0.588 ± 0.040 | 0.567 ± 0.034 | 0.540 ± 0.030 | 0.561 ± 0.018 | 0.518 ± 0.019 | **0.618 ± 0.048** |
| Perindopril MPO | Genetic GFN | **0.588 ± 0.039** | **0.573 ± 0.028** | **0.553 ± 0.025** | **0.517 ± 0.017** | **0.502 ± 0.017** | 0.466 ± 0.122 |
| | MOL GA | 0.582 ± 0.053 | 0.575 ± 0.049 | 0.554 ± 0.045 | 0.526 ± 0.041 | 0.513 ± 0.039 | 0.488 ± 0.024 |
| | Gradient GA | 0.486 ± 0.027 | 0.479 ± 0.026 | 0.462 ± 0.029 | 0.454 ± 0.016 | 0.436 ± 0.018 | **0.572 ± 0.069** |
| Deco Hop | Genetic GFN | 0.641 ± 0.009 | 0.636 ± 0.011 | 0.626 ± 0.011 | 0.609 ± 0.010 | 0.601 ± 0.010 | 0.556 ± 0.084 |
| | MOL GA | **0.649 ± 0.028** | **0.647 ± 0.029** | **0.640 ± 0.027** | **0.619 ± 0.026** | **0.616 ± 0.024** | 0.485 ± 0.032 |
| | Gradient GA | 0.612 ± 0.001 | 0.603 ± 0.005 | 0.595 ± 0.007 | 0.588 ± 0.003 | 0.576 ± 0.005 | **0.644 ± 0.042** |
| Median 1 | Genetic GFN | **0.387 ± 0.030** | **0.365 ± 0.027** | **0.311 ± 0.020** | **0.314 ± 0.027** | **0.291 ± 0.027** | 0.563 ± 0.108 |
| | MOL GA | 0.349 ± 0.031 | 0.325 ± 0.010 | 0.298 ± 0.006 | 0.301 ± 0.015 | 0.279 ± 0.007 | 0.642 ± 0.049 |
| | Gradient GA | 0.318 ± 0.011 | 0.303 ± 0.007 | 0.303 ± 0.009 | 0.258 ± 0.012 | 0.230 ± 0.015 | **0.697 ± 0.037** |
| Isomer c9h10n2o2pf2cl | Genetic GFN | **0.919 ± 0.022** | **0.900 ± 0.024** | **0.858 ± 0.020** | 0.809 ± 0.057 | 0.765 ± 0.059 | 0.671 ± 0.070 |
| | MOL GA | 0.905 ± 0.031 | 0.897 ± 0.021 | 0.855 ± 0.012 | **0.845 ± 0.021** | **0.808 ± 0.019** | 0.735 ± 0.048 |
| | Gradient GA | 0.826 ± 0.094 | 0.778 ± 0.096 | 0.670 ± 0.099 | 0.663 ± 0.073 | 0.544 ± 0.069 | **0.748 ± 0.039** |

Table 17: Comparison of Average Top 1, Average Top 10, Average Top 100, AUC Top 1, AUC Top 10, and AUC Top 100 with several GuacaMol objectives (mestranol similarity, amlodipine MPO, perindopril MPO, deco hop, median1, and isomers c9h10n2o2pf2cl) under 2500 oracle calls.

| Metric | Method | Top 1 | Top 10 | Top 100 | AUC Top 1 | AUC Top 10 | AUC Top 100 | Diversity Top 100 |
|---|---|---|---|---|---|---|---|---|
| Mestranol Similarity | Genetic GFN | 0.7796 ± 0.0724 | 0.7796 ± 0.0724 | **0.7776 ± 0.0695** | **0.7127 ± 0.0443** | **0.7057 ± 0.0420** | **0.6820 ± 0.0337** | 0.2314 ± 0.0901 |
| | MOL GA | **0.8179 ± 0.0974** | **0.8033 ± 0.1019** | 0.7654 ± 0.1018 | 0.6982 ± 0.0889 | 0.6774 ± 0.0864 | 0.6346 ± 0.0771 | 0.4189 ± 0.0442 |
| | Gradient GA | 0.7348 ± 0.0321 | 0.7140 ± 0.0282 | 0.6618 ± 0.0298 | 0.6179 ± 0.0401 | 0.5833 ± 0.0349 | 0.5273 ± 0.0318 | **0.5329 ± 0.0394** |
| Amlodipine MPO | Genetic GFN | 0.7416 ± 0.0658 | 0.7364 ± 0.0356 | 0.7225 ± 0.0613 | 0.6850 ± 0.0457 | 0.6477 ± 0.0518 | 0.6477 ± 0.0518 | 0.4291 ± 0.0715 |
| | MOL GA | **0.8534 ± 0.0238** | **0.8444 ± 0.0216** | **0.8150 ± 0.0165** | **0.7480 ± 0.0158** | **0.7329 ± 0.0130** | **0.6989 ± 0.0099** | 0.4067 ± 0.0338 |
| | Gradient GA | 0.6942 ± 0.0499 | 0.6848 ± 0.0510 | 0.6698 ± 0.0456 | 0.6309 ± 0.0361 | 0.6120 ± 0.0348 | 0.5845 ± 0.0309 | **0.4402 ± 0.0859** |
| Perindopril MPO | Genetic GFN | **0.6213 ± 0.0145** | **0.6178 ± 0.0118** | **0.6108 ± 0.0157** | **0.5812 ± 0.0204** | **0.5765 ± 0.0146** | **0.5593 ± 0.0215** | 0.3871 ± 0.0783 |
| | MOL GA | 0.6166 ± 0.0269 | 0.6095 ± 0.0328 | 0.5947 ± 0.0304 | 0.5699 ± 0.0262 | 0.5635 ± 0.0248 | 0.5444 ± 0.0214 | 0.4523 ± 0.0365 |
| | Gradient GA | 0.5741 ± 0.0104 | 0.5613 ± 0.0109 | 0.5443 ± 0.0113 | 0.5228 ± 0.0113 | 0.5102 ± 0.0110 | 0.4874 ± 0.0111 | **0.5477 ± 0.0594** |
| Deco Hop | Genetic GFN | **0.6976 ± 0.0030** | **0.6958 ± 0.0046** | **0.6900 ± 0.0090** | **0.6626 ± 0.0082** | **0.6569 ± 0.0087** | **0.6458 ± 0.0093** | 0.4323 ± 0.0941 |
| | MOL GA | 0.6734 ± 0.0219 | 0.6713 ± 0.0073 | 0.6689 ± 0.0058 | 0.6347 ± 0.0091 | 0.6263 ± 0.0018 | 0.6218 ± 0.0052 | 0.4196 ± 0.0387 |
| | Gradient GA | 0.6452 ± 0.0101 | 0.6394 ± 0.0088 | 0.6341 ± 0.0085 | 0.6139 ± 0.0067 | 0.6085 ± 0.0066 | 0.6001 ± 0.0061 | **0.4624 ± 0.0515** |
| Median 1 | Genetic GFN | **0.4000 ± 0.0000** | **0.4000 ± 0.0000** | 0.3696 ± 0.0040 | 0.3639 ± 0.0055 | 0.3560 ± 0.0074 | 0.3116 ± 0.0075 | 0.4918 ± 0.0729 |
| | MOL GA | **0.4000 ± 0.0000** | 0.3787 ± 0.0064 | 0.3495 ± 0.0158 | **0.3669 ± 0.0153** | **0.3360 ± 0.0156** | **0.3197 ± 0.0394** | 0.5699 ± 0.0350 |
| | Gradient GA | 0.3923 ± 0.0173 | 0.3714 ± 0.0183 | **0.3378 ± 0.0095** | 0.3376 ± 0.0215 | 0.3108 ± 0.0153 | 0.2789 ± 0.0125 | **0.6191 ± 0.0278** |
| Isomer c9h10n2o2pf2cl | Genetic GFN | **0.9394 ± 0.0000** | 0.9312 ± 0.0153 | 0.9084 ± 0.0476 | **0.9138 ± 0.0649** | **0.8956 ± 0.0813** | **0.8460 ± 0.0525** | 0.7104 ± 0.0362 |
| | MOL GA | 0.9346 ± 0.0084 | **0.9213 ± 0.0265** | **0.9023 ± 0.0237** | 0.9006 ± 0.0210 | 0.8854 ± 0.0207 | 0.8398 ± 0.0116 | 0.7399 ± 0.0732 |
| | Gradient GA | **0.9394 ± 0.0000** | 0.9237 ± 0.0196 | 0.8733 ± 0.0264 | 0.8286 ± 0.0704 | 0.7878 ± 0.0758 | 0.7018 ± 0.0808 | **0.7481 ± 0.0389** |

Table 18: Comparison of Average Top 1, Average Top 10, Average Top 100, AUC Top 1, AUC Top 10, and AUC Top 100 with several GuacaMol objectives (mestranol similarity, amlodipine MPO, perindopril MPO, deco hop, median1, and isomers c9h10n2o2pf2cl) under 10000 oracle calls.

## G  Preliminary Multi-Objective Results

Despite the proposed work being heavily focused on single-objective optimization, we present our future idea about multi-objective optimization in this section. We envision the multi-objective approach for Gradient GAby providing the aggregate of objectives as a single objective to the current framework. We have chosen a simple product as the aggregate function, due to its simplicity and applicability over previous works, such as MARS (Xie et al. (2021)). We have conducted a sample run with 2500 Oracle calls for multi-objective mestranol similarity and amoldipine MPO and have seen comparable results similar to those of single-objective. Tables 19, 20 provide the comparison of the sample run with the original results. This experiment shows that the proposed framework yields similar outcomes for individual objectives even when the target is a joint distribution of the said objectives.

Table 19: mestranol similarity Results: single-objective vs multi-objective

| Method | Average Top 1 | Top 10 | Top 100 | AUC Top 1 | Top 10 | Top 100 | Average SA | Diversity |
|---|---|---|---|---|---|---|---|---|
| Gradient GA | 0.5367±0.0421 | 0.5130±0.0393 | **0.4673±0.0293** | 0.4433±0.0310 | 0.4082±0.0315 | 0.3534±0.0355 | 4.9647±0.3605 | **0.6692±0.0294** |
| Gradient GA(multi-objective) | **0.6528** | **0.5497** | 0.4268 | **0.6398** | **0.5366** | **0.4158** | **4.4780** | 0.6552 |

Table 20: amoldipine MPO Results: single-objective vs multi-objective

| Method | Average Top 1 | Top 10 | Top 100 | AUC Top 1 | Top 10 | Top 100 | Average SA | Diversity |
|---|---|---|---|---|---|---|---|---|
| Gradient GA | **0.5880±0.0397** | **0.5667±0.0336** | **0.5400±0.0296** | **0.5614±0.0177** | **0.5176±0.0187** | 0.4658±0.0199 | **3.8677±0.3122** | **0.6176±0.0478** |
| Gradient GA(multi-objective) | 0.5641 | 0.5083 | 0.4860 | 0.5170 | 0.4937 | **0.4738** | 4.2944 | 0.5476 |

