# OpenReview forum: "Gradient GA: Gradient Genetic Algorithm For Drug Molecular Design"
_TMLR — Accepted by TMLR_

### Review · Reviewer_vDAC · 2025-08-15

**Summary Of Contributions:**

This paper proposes a gradient genetic algorithm for drug molecular design. The algorithm learns a differentiable objective function with a GNN to map the graph-structured information of molecules to vector embeddings. Then it applies discrete langevin proposal to incorporate gradient information from the objective, which accelerates convergence concerning tranditional genetic search.

**Audience:**

Yes

**Audience Explanation:**

Researchers from the molecular design and evolutionary algorithm communities may be interested in the paper.

**Claims And Evidence:**

Yes

**Claims Explanation:**

The claims are well described and supported by experimental results.

**Requested Changes:**

1. The gradients are derived from an NN surrogate trained to approximate the oracle. If the surrogate is inaccurate, especially in early iterations or in noisy oracle settings, the gradient direction can be misleading. The paper would benefit from a robustness analysis on surrogate error and its impact on optimization performance.
2. One significant advantage of genetic algorithm is its diversity discovery ability. While gradient-based search helps fine-grained local exploitation, it risks overfitting to a narrow region of the search space and lossing solution diversity. How can the algoirthm make a balance between onvergence and diversity?
3. Similiar to point 2, only the best-scoring parent contributes to the gradient update, discarding information from the other parent. This limits the potential benefits of crossover and may reduce structural innovation. Exploring gradient aggregation from both parents could improve exploration.
4. The algorithm requires frequent retraining of the surrogate and gradient computations. Although the number of oracle calls is capped, the paper does not discuss the additional computational overhead from model training and whether it remains practical in more expensive real-world scenarios.

---

> ### Author Response · Authors · 2025-09-14
>
> We would like to thank the reviewer for their insightful comments. Below are our responses to the reviewer’s concerns:
> 1. We agree that if the surrogate were inaccurate, the gradient direction could be misleading. However, our results indicate that the **surrogate is indeed a reliable approximation of the oracle**, as evidenced by the **strong benchmark performance** across diverse tasks. To further strengthen this point, we will include an empirical analysis by withholding part of the training data to create a validation set, explicitly quantifying surrogate error. We will report these results once obtained.
> 2. We acknowledge that gradient-based search naturally emphasizes exploitation over exploration. Nevertheless, our method achieves a **favorable balance compared to existing approaches**: in Appendix A.2, Graph GA achieved a diversity of 0.56, our method 0.53, while the RL-based method, REINVENT, dropped to 0.30. This demonstrates that our approach maintains **significantly higher diversity than RL-based baselines** while remaining competitive with GA. Importantly, our algorithm provides a **tunable hyperparameter** $\alpha$ in the DLP formula that explicitly controls this tradeoff. For example, in a 1000-call run, $\alpha=1$ yielded diversity of 0.59 with a top-1 score of 0.6, while $\alpha=1000$ yielded diversity of 0.73 with a top-1 score of 0.5. This built-in flexibility allows practitioners to adaptively balance convergence and diversity depending on the application.
> 3. While we currently use the best-scoring parent, the offspring also inherit fragments from the other parent, preserving crossover benefits. Furthermore, we **experimented with averaging the scores of both parents**: while this increased diversity (0.59 → 0.70), it reduced top-k performance (0.60 → 0.53). Thus, prioritizing the best parent yields higher-quality solutions, whereas incorporating both parents improves diversity. We will clarify this design choice and include these results in the revision.
> 4. We acknowledge that gradient computation introduces additional overhead compared to standard GA. However, **the cost remains modest and practical**: for a 1000-call run, GA required about 1 minute, while our method required 2-3 minutes. In contrast, RL-based methods required 4-5 minutes, making our approach about **0.6×** their runtime. Thus, while slightly more expensive than GA, our method remains significantly more efficient than RL-based baselines while achieving superior diversity and optimization tradeoffs. We will expand the discussion of these results to highlight practical applicability in real-world scenarios.

---

> > ### Comment · Reviewer_vDAC · 2025-09-15
> > **Reviewer's rely to the authors'response**
> >
> > Thanks for the authors' response. The response makes sense to my previous comments. I don't have further concerns.

---

> > > ### Author Response · Authors · 2025-09-26
> > >
> > > We have performed an analysis of the surrogate error across all training calls for the neural network. Across 10000 oracle calls, there are a total of 101 calls for training the network, including training for the initial sample. We have considered two different validation sets: (i) a validation set from the training set, (ii) a validation set from the current population set. The size of the validation set is considered 10% of their respective total sets. We have calculated the total training loss, total validation loss from the training set, and total validation loss from the known samples, and added our analysis in Appendix D. We have added Figure 12 in the manuscript, plotting all these losses over the number of oracle calls, and it shows that the initial validation losses are not that significant at maximum 0.001 and decrease over time. This shows the surrogate is an accurate approximation.

---

### Review · Reviewer_Ttib · 2025-08-18

**Summary Of Contributions:**

Summary:
This paper presents Gradient GA, a new genetic algorithm for drug molecular design that uses gradient guidance from neural network-based objectives to efficiently explore chemical space. Traditional genetic algorithms operate by random operations, which can be slow and inefficient. Gradient GA replaces random changes with informed moves guided by gradients in a differentiable space, made possible by a message passing neural network (MPNN) that predicts molecular properties. The approach demonstrates faster convergence and higher-quality molecule generation across several benchmark tasks when compared against state-of-the-art optimization methods like Graph GA, SMILES-GA, MIMOSA, MARS, and DST.

Strength:
The integration of gradient information into a GA framework for molecular discovery is innovative and clearly explained. The manuscript presents extensive comparative data across multiple baselines and metrics, notably using the number of oracle calls. Results span multiple molecule optimization tasks and metrics (Average Top-K, AUC Top-K, diversity, and synthetic accessibility) against other algorithms. Proves the benchmarks by showing fewer oracle calls needed for molecular property evaluation, compared to other algorithms.

Weakness:
Gradient GA relies heavily on neural embeddings to represent molecules for optimization. However, the paper does not fully address how the embedding space is defined for multiple features or objectives. If embeddings for different molecular features are not aligned or mapped consistently, the optimization process may become biased and less effective, especially for multi-objective molecular design. This technical gap could lead to gradients that do not truly guide the search towards molecules that are optimal across all desired properties.

**Audience:**

Yes

**Audience Explanation:**

Gradient GA will be valuable for researchers in computational chemistry, drug discovery, and machine learning for scientific applications. Specifically, those working on molecular optimization, virtual screening, graph neural networks, and efficient algorithmic search will benefit from these results

**Broader Impact Concerns:**

.

**Claims And Evidence:**

Yes

**Claims Explanation:**

The paper's claims are strongly supported by quantitative data. Improvements in convergence speed and solution quality are shown through direct comparisons of AUC Top-K scores versus the number of oracle calls. Tables and figures illustrate that Gradient GA requires fewer expensive oracle calls to reach superior results on common molecular benchmarks compared to other algorithms.

**Requested Changes:**

Clarifying whether embeddings for distinct objectives are jointly normalized or aligned, and how gradients are mapped and interpreted when objectives have different semantics or scales, would increase the transparency of the method and guide future extensions to multi-objective tasks. Even a brief note on this point, such as discussing embedding coordination or normalization across features, would not require major changes to the algorithm design, yet would augment the paper’s clarity and its value to researchers working on general-purpose molecular optimization.

---

> ### Author Response · Authors · 2025-09-14
>
> We would like to thank the reviewer for their insightful comments. Below are our responses to the reviewer’s concerns:
>
> This paper focuses on  **single-objective** tasks, but our method naturally extends to multi-objective settings, which we agree is an interesting direction for future work. We will provide preliminary results for a simple multi-objective setup in the coming days.
>
> Regarding embeddings, our approach does not modify the parent embeddings produced by the MPNN; these embeddings are  **learned directly** from molecular structures, ensuring that the latent space is consistently defined. During MPNN retraining, we normalize population scores at the end of each step so that they fall within [0,1]. In addition, we scale the gradient with respect to the objective value by dividing it by the parent scores, as shown in Eq. (6). We believe these  **normalization and scaling mechanisms** will help balance multiple objectives and prevent bias when objectives have different magnitudes.
>
> In the benchmarks considered in the paper, all oracle values already lie within [0,1]. If oracles had different scales, they could be rescaled in a manner similar to MARS [1], which adjusted the SA objective from [0,10] to [0,1] to ensure compatibility with other objectives.
>
>
> [1] Xie, Yutong, et al. "MARS: Markov Molecular Sampling for Multi-objective Drug Discovery." International Conference on Learning Representations (2021).

---

> > ### Author Response · Authors · 2025-09-26
> >
> > We have performed a sample experiment on multi-objective optimization, envisioning our future work. In order to perform multi-objective optimization in our current framework, we have used the product of all objectives as the final objective. The experiment settings have been conducted for 2500 oracle calls for the objectives mestranol similarity, amoldipine MPO. The results and analysis wrt current work results have been detailed in Table 19, Table 20 of Appendix G in the updated manuscript. The results show that multi-objective optimized molecules produce similar results to the single objective. This study shows that the proposed framework, without any fundamental changes, can generate aligned embeddings for multiple objectives.

---

> > ### Comment · Reviewer_Ttib · 2025-09-26
> >
> > Thanks for extending to multi-objective. I don't have any further comments.
> > Thank you!

---

### Review · Reviewer_DK6J · 2025-08-29

**Summary Of Contributions:**

In this paper, the authors introduce Gradient GA, a genetic algorithm for molecular graph generation that includes gradient information to guide the generation process. To handle the difficulty linked to the discrete nature of graphs, the authors leverage an embedding method with a GNN. The graphs are embedded in some latent space, these representations are used to approximate an oracle objective function on molecules, and this latter part is used to compute gradients from the latent space to the objective. This gradient is then used to guide the sampling process, akin to Discrete Langevin. The authors report good experimental results on several datasets, with several classical metrics in molecular graph generation.

**Audience:**

Yes

**Audience Explanation:**

Molecular graph generation is a very active area of research, and primary field of application, for graph generative models.

**Claims And Evidence:**

Yes

**Claims Explanation:**

The paper is clear, and the claims are convincing.

**Requested Changes:**

I found the paper quite clear and well-written, and I generally like the proposed method, which I find clever and well-put together.

I may have minor remarks concerning notations; it seems that some notations may be redundant and/or slightly useless, which results in a multiplication of them that may obfuscate a bit the main ideas, underselling one strength of the proposed method, which is to be quite simple and straightforward (the authors even give a table of notations, which I find a tad too much here). For instance, the ``objective'' function/mapping appears under many form: $\mathcal{O}$, $f$, $L_2$, $\mathcal{M}$... and it can be a bit tedious to identify that some of those are the same object. It may be possible to gain clarity by being a bit more straighforward here.

---

> ### Author Response · Authors · 2025-09-13
>
> We would like to thank the reviewer for such meticulous comments. We will clean up the redundant/rarely-used notations to make it easier to follow. At the same time, the use of different notations was because the underlying formulations are not the same, and we aim to clearly connect the different backgrounds. We will revise the presentation to balance clarity with mathematical rigor, making sure that any differences in notation are used only where they truly help convey nuance.

---

### Author Response · Authors · 2025-09-26
**Summary Response**

We thank all reviewers for their constructive comments and suggestions, which have helped us improve both the quality and presentation of the paper. We have updated the PDF accordingly (changes are in blue), with the main changes summarized below:
- Revised the notation table by removing hyperparameters and retaining only the key notations.
- Added a short discussion on real-world applications in the conclusion and future work section.
- Included the surrogate error loss analysis in Appendix D.
- Included the parent selection results in Appendix E.
- Added multi-objective test results in Appendix G.

---

### Decision · Action_Editor_5qPP · 2025-10-23

**Recommendation:** Accept as is

**Audience:**

Yes

**Audience Explanation:**

The topic is relevant to graph neural networks and graph generative models, which is an active topic of research.

**Claims And Evidence:**

Yes

**Claims Explanation:**

The paper is clear and its experimental setup is convincing and supports the claims made in the paper.